# AddressVLM: Cross-view Alignment Tuning for Image Address Localization using Large Vision-Language Models

## Abstract

Large visual language models (LVLMs) have demonstrated impressive performance in coarse-grained geo-localization at the country or city level, but they struggle with fine-grained street-level localization within urban areas. In this paper, we explore integrating city-wide address localization capabilities into LVLMs, facilitating flexible address-related question answering using street-view images. A key challenge is that the street-view visual question-and-answer (VQA) data provides only microscopic visual cues, leading to subpar performance in fine-tuned models. To tackle this issue, we incorporate perspective-invariant satellite images as macro cues and propose *cross-view alignment tuning* including a satellite-view and street-view image grafting mechanism, along with an automatic alignment label generation mechanism. This helps build connections between street-view images through cross-view matching, thus enhancing LVLM's global understanding of street distribution. We name our proposed model *AddressVLM* consisting of two-stage training protocols: cross-view alignment tuning and address localization tuning. Furthermore, we have constructed two street-view VQA datasets based on image address localization datasets from Pittsburgh and San Francisco. Qualitative and quantitative evaluations demonstrate that AddressVLM outperforms counterpart LVLMs by over 9% and 12% in average address localization accuracy on the Pitts-VQA and SF-Base-VQA datasets, respectively.

## 1 Introduction

Visual place recognition (VPR) aims to predict the geographic location of a given image, which can be categorized into two types: image geo-localization (Arandjelovic et al., 2016; Wang et al., 2022; Ali-Bey et al., 2023) and image address localization (Xu et al., 2024). The emergence of Large Vision-Language Models (LVLMs), such as GPT-4V (Achiam et al., 2023), Qwen-VL (Bai et al., 2023), and LLaVA (Liu et al., 2024), have significantly impacted various tasks related to images and languages. As generative models capable of generating natural language, they demonstrate enhanced adaptability and flexibility in the image localization task (Yang et al., 2023). This proficiency stems from the extensive exposure to street-view and landmark images during their training phases.

Recent work, GeoReasoner (Li et al., 2024), integrates a large vision-language model with human inference knowledge for street view geo-localization with reasoning, presenting significant advantages in coarse-grained localization at the country or city level. However, when it comes to address localization for specific districts (*i.e.*, Downtown) or streets (*i.e.*, Fifth Avenue) within a city, it may struggle to predict accurate textual address, since street-view images are more similar and difficult to distinguish and the street-level address names have not been adequately correlated with the corresponding street-view images. In contrast, the previous work AddressCLIP (Xu et al., 2024) explores city-wide address localization by contrastive learning between street-view images and textual address. Nevertheless, this approach is inherently limited due to its reliance on a discriminative model that can only make distinctions among a constrained set of candidate addresses. As a result, it lacks the flexibility to provide versatile address descriptions and answer other related inquiries.

To combine the advantages of previous work, in this study, we explore how to integrate street-level address localization capabilities into an LVLM. The model is expected to respond flexibly to user

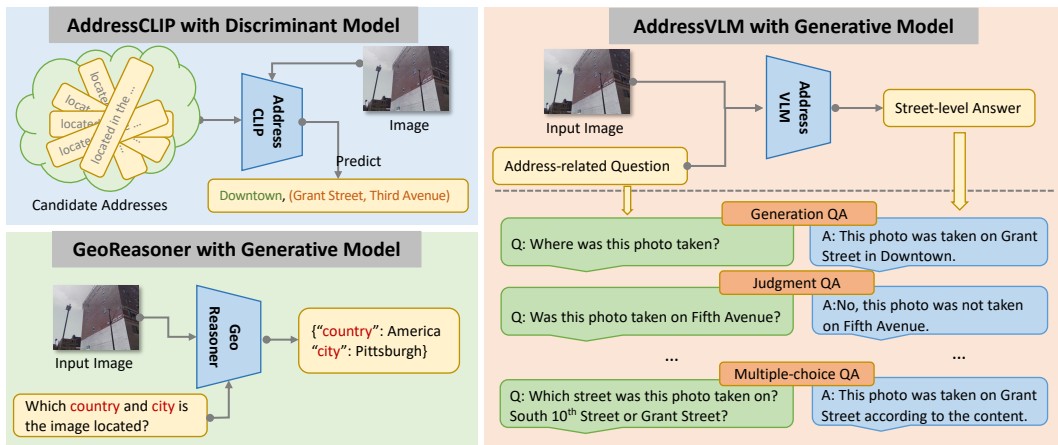

Figure 1: Comparison of our AddressVLM with AddressCLIP and GeoReasoner. Our approach focuses on city-wide image address localization and flexible address questions and answers related to address using large vision-language models.

inquiries about address localization. We name our model *AddressVLM*, which is designed to handle address-related questions and provide answers accurate to the district and street level. Fig. 1 shows the comparisons of the proposed AddressVLM with AddressCLIP and GeoReasoner. Our method can answer various types of questions including generation, judgment, and multiple-choice.

To realize the above goal, a reasonable approach involves fine-tuning a well-trained LVLM using street-view question-and-answer (VQA) data with LoRA adaptation (Hu et al., 2021). However, this straightforward method of *address localization tuning* yields suboptimal performance. The primary reason is that street-view images are sparsely collected in terms of both location and viewpoint, which inhibits the model's ability to build a global understanding of street distribution across an entire city. Such global information is crucial for effective address localization since street-view images are densely sampled during testing. To supplement the global information in fine-tuning, we introduce perspective-invariant satellite images to establish connections between sparse street-view images. Satellite images are globally consistent and exhibit overlap, allowing for a mapping of the sparse street-view images to a global framework that facilitates inter-image correlations.

Previous research in cross-view geo-localization (Durgam et al., 2024) has shown the viability of correlating satellite images with street-view images. In light of this, we propose a method named *cross-view alignment tuning*, designed to enable LVLMs to align street-view images with street addresses on satellite images annotated with street name labels. This method integrates a global understanding of street distributions within urban environments into LVLMs. It consists of two key components: the satellite-view and street-view image grafting mechanism and the automatic alignment label generation mechanism. The former places street-view images in the upper right corner of their corresponding regional satellite images, serving as the input for cross-view alignment tuning. The latter employs an off-the-shelf LVLM to explain why the street-view image matches the address in the satellite images according to the provided address hint, thus automatically generating labels for the cross-view alignment tuning. By doing this, our full method involves two-stage training protocols: cross-view alignment tuning and address localization tuning.

We introduce two city-wide street-view VQA datasets named Pitts-VQA and SF-Base-VQA, built upon the Pitts-IAL (Torii et al., 2013; Xu et al., 2024) and SF-Base-IAL (Berton et al., 2022; Xu et al., 2024) datasets, respectively. On Pitts-VQA, AddressVLM demonstrates an improvement of 9% compared to the baseline without cross-view alignment tuning. On SF-Base-VQA, AddressVLM achieves an improvement of 12% over the baseline. Moreover, in comparison to the state-of-the-art (SOTA) approach for image address localization using LVLMs, GeoReasoner (Li et al., 2024), our method exhibits improvements of 11% and 14% on the Pitts-VQA and SF-Base-VQA datasets, respectively. The proposed method exhibits excellent city-wide address localization capability compared to general LVLMs. We further provide qualitative results to thoroughly validate the effectiveness of the proposed cross-view alignment tuning strategy. Additional quantitative experiments show that our method can be extended to address localization in multiple cities.

Overall, the main contributions of our work are summarized as follows:

- We explore integrating city-wide address localization capabilities into LVLMs to enable flexible address question and answer based on street-view images.

- We introduce cross-view alignment tuning that integrates the global understanding of urban street distribution into LVLMs, which includes the satellite-view and street-view image grafting mechanism and the automatic alignment label generation mechanism.

- We propose AddressVLM, an LVLM that achieves consistent improvements over the baseline without cross-view alignment tuning on street-view VQA datasets and performs superior to the SOTA method GeoReasoner and general LVLMs.

## 2 RELATED WORK

**Visual Place Recognition.** Visual place recognition aims to predict the geographic location of a given image with broad applications in practical scenarios (Zhang et al., 2021). Most researchers have focused on predicting the latitude and longitude coordinates for input images, known as image geo-localization, which is primarily categorized into retrieval-based methods (Hausler et al., 2021; Wang et al., 2022; Ali-Bey et al., 2023; Keetha et al., 2023) and classification-based methods (Seo et al., 2018; Pramanick et al., 2022; Clark et al., 2023; Trivigno et al., 2023). Retrieval-based methods involve matching the given image with a database of images tagged with GPS coordinates and retrieving the geographical coordinates of the most similar images as the prediction result. Classification-based methods, on the other hand, subdivide the Earth's surface or cities into thousands of geographical cells and predict the geographical unit to which an image belongs. Recent trends have involved leveraging the general text knowledge embedded in visual-language models for geo-localization, including CLIP-based (Radford et al., 2021) discriminative models such as Street-CLIP (Haas et al., 2023) with region descriptions and GeoCLIP Cepeda et al. (2023) with GPS information injection, as well as LVLM-based generative models like GeoReasoner (Li et al., 2024) with human reasoning knowledge. However, these models typically focus only on coarse-grained localization at the country or city level. Recent efforts represented by AddressCLIP (Xu et al., 2024) focus on fine-grained street-level localization within a city, yet this discriminative model is constrained to make distinctions within a limited set of candidate addresses and cannot provide flexible address descriptions or question-and-answer as generative models can. In this study, we explore integrating fine-grained city-wide address localization capability into LVLMs.

**Large Vision Language Models.** LVLM has been a new rising research hotspot, which uses powerful Large Language Models (LLMs) (Touvron et al., 2023; Jiang et al., 2023; Yang et al., 2024; Abdin et al., 2024) as a brain to perform vision-language tasks. These general-purpose LVLMs exhibit remarkable effectiveness in visual question-answering tasks (Achiam et al., 2023; Bai et al., 2023; Liu et al., 2024; Team et al., 2023), suggesting a potential path to artificial general intelligence. For VPR, LVLMs can identify the location of input images based on landmarks, Optical Character Recognition (OCR) information, or other notable visual cues, often achieving precision at the level of country or even city (Yang et al., 2023). However, the optimal utilization of LVLMs for fine-grained street-level localization remains a challenging issue. This study leverages the capabilities of LVLMs to tackle image address localization in street views. We overcome these challenges through cross-view alignment tuning by introducing satellite images from a macro perspective, thus contributing to a more effective application of LVLMs in this domain.

**Cross-view Geo-localization.** The objective of cross-view geo-localization is similar to VPR, except that its database consists of aerial images instead of ground street views, and the queries might be panorama images. The key challenge is to match features between aerial and ground images in the feature space (Durgam et al., 2024). A classic approach to tackle this issue is the implementation of Siamese networks for alignment, as suggested by Vigor (Zhu et al., 2021). To address temporal changes in ground images, the authors in (Ghanem et al., 2023) focus on the temporally invariant parts of images. Additionally, some work (Wang et al., 2021; Mi et al., 2024) propose part-based image representation learning to address the orientation and local detail matching issues. Overall, these studies demonstrate the potential for correlating aerial images with street-view images. Inspired by the spirit of cross-view matching, we apply this task to the domain of LVLMs and adapt it to introduce the method of cross-view alignment tuning.

## 3 METHOD

### 3.1 PROBLEM STATEMENT

The Image Address Localization problem with Visual Question Answering is formalized as follows: given a training dataset $D_{train} = \{(I_i, Q_i^j, A_i^j)\}_{i=1}^M, j \in [1...N_i]$, where $I_i$ represents images and $(Q_i^j, A_i^j)$ denotes multi-turn questions and answers, our objective is to train a large vision-language model $\mathcal{H}_\theta$ to predict answers based on the query images and address-related questions. During the training phase, for each image $I_i$, we organize the multi-turn conversation data as a sequence, by treating all answers as the model's response, and the instruction $S_i^t$ at the $t-$th turn as:

$$S_i^t = \begin{cases} [I_i, Q_i^1], t = 1 \\ \quad Q_i^t, t > 1 \end{cases}.$$ (1)

We perform address localization tuning of the LLM on the prediction tokens, using its original autoregressive training objective. Specifically, for a sequence of length $N$, we compute the probability of the target answers $A_i$ by:

$$p(A_i|I_i, S_i) = \prod_{j=1}^N p_\theta(x_j|I_i, S_{<j}, A_{<j}),$$ (2)

where $\theta$ is the trainable model parameters, $S_{<j}$ and $A_{<j}$ are the instruction and answer tokens in all turns before the current prediction token $x_j$, respectively. In the testing phase, given a query image $I_k$ and a set of relevant dialogue questions $Q_j^k$, the model aims to output the corresponding answers $A_j^k$ for each question. The final output consists of natural language responses that provide relevant information regarding the image address, effectively enabling the model to handle the visual question-answering task in the context of address localization.

### 3.2 CROSS-VIEW ALIGNMENT TUNING

Street-view images, serving as sparse micro-level visual cues, make it challenging to provide the model with a global macro perspective, which is crucial for effective address localization since street-view images are densely sampled during testing. In contrast, satellite images can be regarded as supplementary macro information, which are perspective-invariant and globally stable to establish connections between sparse street-view images. Inspired by previous works of cross-view matching (Durgam et al., 2024; Hao et al., 2024), we propose cross-view alignment tuning to align the street-view images with the corresponding street address on satellite images. This helps LVLMs first to achieve a global understanding of the spatial distribution of urban streets and then to build a fine-grained understanding of image-address matching.

**Satellite-view and Street-view Image Grafting.** To align satellite-view and street-view images, two intuitive approaches can be considered: i) directly concatenating the two images into a single input, and ii) treating the two images as separate inputs. In both approaches, the equal contribution of the two images can dilute the effectiveness of satellite images. Furthermore, modern training techniques for LVLMs usually resize images to a uniform size, which can lead to substantial distortion when directly stitching the two images together. While some studies have investigated input structures that accommodate dual images (Wu et al., 2024), the second approach diverges from the mainstream LVLM architectures (Achiam et al., 2023; Bai et al., 2023; Liu et al., 2024).

To address the aforementioned issues, we propose a satellite-view and street-view image grafting mechanism, where street-view images are scaled down and grafted onto satellite images like CutMix data augmentation (Yun et al., 2019). Let $I_{sa}$ and $I_{st}$ denote the satellite image and street-view image, respectively. The grafting goal is to generate a new image $I_s$ by combining the two view images. The grafting operation can be expressed as:

$$I_s = \mathbf{M} \odot I_{sa} + (\mathbf{1} - \mathbf{M}) \odot I_{st},$$ (3)

where $\mathbf{M}$ denotes a binary mask indicating where to drop out and fill in from two view images, $\mathbf{1}$ is a binary mask filled with ones, and $\odot$ is element-wise multiplication. According to cartographic and visualization conventions, we position the street-view image in the upper right corner of the satellite

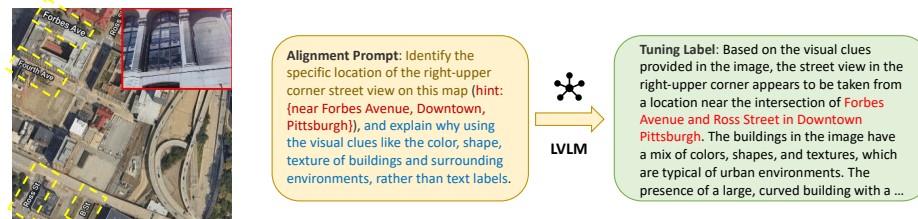

(a) Satellite and Street-view Image Grafting      (b) Automatic Alignment Label Generation

Figure 2: Schematic diagram of satellite and street-view image grafting (a), as well as an example of the alignment prompt and generated label for automatic alignment label generation (b). The red and yellow boxes in (a) are only for highlighting and are not marked in the fine-tuning data.

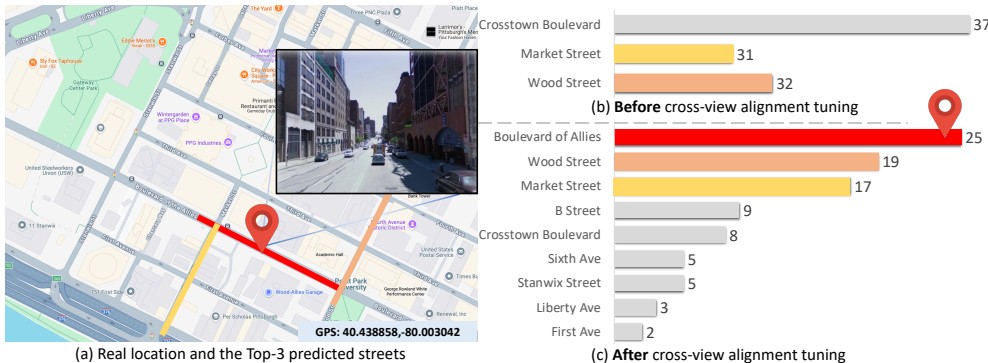

(a) Real location and the Top-3 predicted streets      (c) **After** cross-view alignment tuning

Figure 3: Qualitative comparisons of the street localization probability distribution before and after cross-view alignment tuning. The predicted streets are clustered and distributed close to the true location after cross-view alignment tuning. The source map can be found here.

image, ensuring a longer side overlap ratio $\delta \in [0, 0.5]$, as shown in Fig. 2. It is worth noting that the text name of each street is marked on the satellite image, which facilitates the alignment of street-view images and street addresses. This mechanism allows a single image to be used as input, where the satellite image serves as the primary focus and the street-view image acts as a supporting element. This setup creates a framework for understanding the relationship between the two images. We analyze the effects of different grafting parameters by ablation experiments in Sec. 4.3.

**Automatic Alignment Label Generation.**

To enable LVLMs to establish a global understanding of urban street layouts using maps, we design a cross-view alignment tuning task. This task allows the model to locate the address of a street-view image by visually matching it with satellite images, where the corresponding textual street name is marked. Meanwhile, we require the model to give the reason for the address prediction. During performing the cross-view alignment tuning task, the model can perceive surrounding street information since LVLMs have a certain OCR capability.

The goal of alignment tuning relies on training the model with appropriate textual labels. An intuitive way is to construct textual labels based on artificial rules and template languages, but this way cannot achieve flexible and diverse language descriptions. To this end, we propose an automatic alignment label generation mechanism. In this mechanism, reference answers based on rules are given in advance, and the reasons are predicted by a well-trained LVLM as textual labels according to reference answers. Here, we provide a *text hint* in the alignment prompt as the standard answer to help generate tuning labels. Fig. 2 shows the pipeline of automatic alignment label generation mechanism with the prompt of label generation. Then, the reference answers are hidden and the alignment tuning is performed using the generated labels.

**Discussion.** To demonstrate the effectiveness of the proposed cross-view alignment tuning, we provide qualitative comparisons of the street localization probability distribution before and after the alignment tuning as shown in Fig. 3. Specifically, we set the temperature parameter of LLM to 0.8 to increase inference variability. Then we perform model inference 100 times for each input street-view

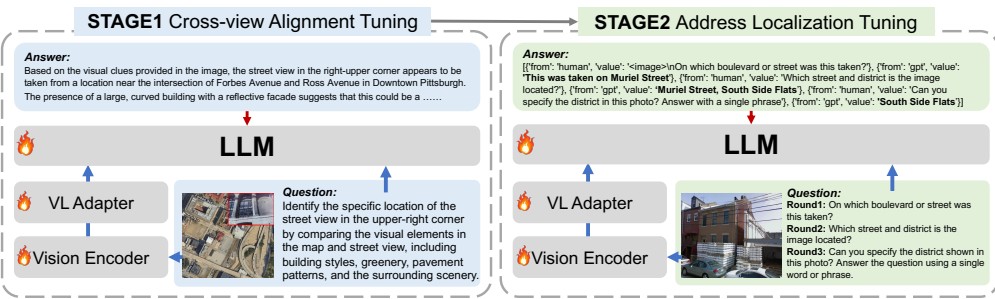

Figure 4: Overview of the proposed framework, which consists of two-stage training protocols: cross-view alignment tuning and address Localization tuning.

image with the specific prompt (*i.e.*, *identify the specific location of the street-view image*). For each sample, we record the frequency of each street appearing in the 100 inference results to approximate the model's understanding of the surrounding street distribution before and after the cross-view alignment tuning. The red marker on the road map indicates the ground truth location of the input image, and the highlighted streets are the Top-3 most frequent outputs. It can be observed that the predicted streets are clustered and distributed close to the ground truth location after cross-view alignment tuning, indicating that the proposed tuning strategy successfully integrates the knowledge of urban street distribution with LVLMs.

## 3.3 TWO-STAGE TRAINING PROTOCOLS

**Street-View Visual Question-and-Answer Datasets.** Due to the absence of a dedicated Visual Question Answering (VQA) dataset specifically for image address localization, we have constructed two street-view VQA datasets tailored for address-related question answering (QA). These datasets are based on image address localization datasets sourced from Pittsburgh (Torii et al., 2013; Xu et al., 2024) and San Francisco (Berton et al., 2022; Xu et al., 2024). To enrich the diversity of the QA data, we have conceived three distinct address QA modes: *generation*, *judgment*, and *multiple-choice*. The generation mode requires the model to answer the accurate address of the location where the input image was taken. The judgment mode requires the model to judge whether the address in the question is correct. The multiple-choice mode requires the model to select the correct address among a given set of addresses. The QA data is generated automatically using language templates and is organized through a series of multiple dialogue rounds. We have designated the VQA datasets corresponding to these two cities as Pitts-VQA and SF-Base-VQA. Specifically, Pitts-VQA contains 10,586 locations with 24 images from different viewpoints for each location and 7 rounds of QA for each image. SF-Base-VQA contains 17,067 locations with 12 images from different viewpoints for each location and 7 rounds of QA for each image. Both datasets are divided into training sets, validation sets, and test sets in a ratio of 7:2:1. We will release these two street-view VQA datasets to the community to promote the research of image address localization.

**Model Architecture.** Fig. 4 illustrates the architecture of the proposed AddressVLM model, designed based on the framework established by LLaVA (Liu et al., 2024). The model consists of three modules: the Vision Encoder $g$, the Vision-Language (VL) Adapter $h$, and the Pre-trained LLM $f$. For an input satellite-view or street-view image $I$, the Vision Encoder with a Vision Transformer (ViT) architecture provides the visual feature $Z_v = g(I)$. The VL Adapter implemented by an MLP layer maps the visual features into language embedding tokens, expressed as $H_v = h(Z_v)$, where $H_v \in \mathbb{R}^{N \times D}$ represents refined visual features that are compatible with textual representations. For another input of textual address query $Q$, we obtain the embedded tokens from the address query as $T_v = \Theta(Q)$, where $\Theta$ represents the off-the-shelf Tokenizer and Embedding models. Finally, the compressed visual feature sequence and the text sequence are concatenated to feed into the Pre-trained LLM module, represented as $A = f(H_v, T_v)$.

**Supervised Fine-tuning.** The overall model undergoes a staged pre-training process that is divided into two phases: cross-view alignment tuning and address localization tuning. In the first stage, our objective is to integrate the spatial distribution of streets and districts within the entire city into LVLMs through the matching between satellite-view images and street-view images for address

Table 1: Performance comparisons with other address localization methods on the Pitts-VQA and SF-Base-VQA datasets.

| | Method | District | | | | Street | | | | $\bar{A}$ | $A_{ds}$ |
|---|---|---|---|---|---|---|---|---|---|---|---|
| | | $A_d^G$ | $A_d^J$ | $A_d^M$ | $\bar{A}_d$ | $A_s^G$ | $A_s^J$ | $A_s^M$ | $\bar{A}_s$ | | |
| Pitts-VQA | AddressCLIP | - | - | - | - | - | - | - | - | - | 82.62 |
| | LLaVA-Phi3-mini | 26.64 | 60.22 | 37.81 | 45.52 | 0.00 | 56.23 | 34.62 | 36.69 | 41.01 | 0.00 |
| | Baseline | 84.51 | 92.72 | 93.23 | 90.70 | 64.31 | 90.25 | 91.27 | 84.00 | 87.27 | 60.52 |
| | GeoReasoner | 83.29 | 91.65 | 91.50 | 89.41 | 61.89 | 89.87 | 89.68 | 82.80 | 86.03 | 57.78 |
| | AddressVLM (ours) | **88.73** | **93.54** | **95.16** | **92.70** | **72.51** | **91.70** | **93.98** | **87.46** | **90.02** | **69.60** |
| SF-Base-VQA | AddressCLIP | - | - | - | - | - | - | - | - | - | 87.44 |
| | LLaVA-Phi3-mini | 3.78 | 71.73 | 42.76 | 46.89 | 0.15 | 52.39 | 30.85 | 33.85 | 40.31 | 0.00 |
| | Baseline | 82.19 | 93.46 | 93.14 | 90.49 | 65.48 | 88.25 | 88.57 | 82.61 | 86.51 | 58.62 |
| | GeoReasoner | 81.40 | 91.07 | 90.81 | 88.53 | 62.89 | 86.46 | 84.64 | 80.08 | 84.26 | 55.99 |
| | AddressVLM (ours) | **86.48** | **93.72** | **94.50** | **92.06** | **76.09** | **88.92** | **92.75** | **86.66** | **89.33** | **70.45** |

localization. This alignment tuning procedure is vital for facilitating the second stage of address localization. In the second stage, we integrate the global prior knowledge of street distribution information to infer the fine-grained, city-wide address location information. Here, we utilize the street-view (VQA) data for the second stage tuning without satellite-view images. Both stages are fine-tuned from the pre-trained LLM using Low-Rank Adaptation (LoRA), which contributes to the overall performance improvements in address localization. This two-stage approach allows the model to better capture complex relationships within the image-address pairs, enhancing its ability to localize addresses accurately by leveraging integrated spatial knowledge.

# 4 EXPERIMENTS

## 4.1 EXPERIMENTAL SETUP

**Implementation Details.** AddressVLM is built upon CLIP (Radford et al., 2021) and Phi-3.1-mini (Abdin et al., 2024) in a LLaVA fashion using the xtuner (Contributors, 2023) framework, which is implemented with PyTorch. All images are adjusted to 336×336 to fit the input size of the CLIP. More details are provided in Appendix A.

**Evaluation Metrics.** To rigorously assess the model's address localization capabilities across diverse conversational contexts, we employ various formats and metrics to assess different levels of localization accuracy. Specifically, we formulate three types of questions: generation, judgment, and multiple-choice. These three measurement formats are applied at both district and street levels. We denote the accuracy for Generation, Judgment, and Multiple-choice question related to district as $A_d^G$, $A_d^J$, and $A_d^M$, respectively, with their average represented as $\bar{A}_d$. Correspondingly, the accuracies for street-level assessments are denoted as $A_s^G$, $A_s^J$, $A_s^M$, with an average of $\bar{A}_s$. The overall accuracy of both district and street level localization is represented as $\bar{A}$. In addition, we investigate the model's capability to concurrently generate both street and district information, referred to as $A_{sd}$. This metric shares some resemblance to the street-level top-1 accuracy (SA-1) in AddressCLIP (Xu et al., 2024). However, it is worth noting that the $A_{sd}$ we report pertains to generative models, making it a more challenging measure than the discriminative SA-1.

## 4.2 MAIN RESULTS

**Baselines.** First, we evaluate the adopted pre-trained LVLM on the metrics above to evaluate its original capabilities in image address localization, which is denoted by **LLaVA-Phi3-mini**. Subsequently, we reproduce the results of **GeoReasoner** (Li et al., 2024) at the district and street levels within a single city as the SOTA method. More method details can be found in Appendix D. Additionally, we conduct only address localization tuning on LLaVA-Phi3-mini, and this tuned model is referred to as **Baseline** for both GeoReasoner and our AddressVLM. Moreover, we compare the street-level results with AddressCLIP with $A_{sd}$ only for completeness.

**Comparisons.** Tab. 1 shows the results of our AddressVLM and the aforementioned models on the Pitts-VQA and SF-Base-VQA datasets. Focusing exclusively on generative models, our approach

Table 2: Ablation study of grafting overlap ratio $\delta$ and satellite image type for cross-view alignment tuning on the Pitts-VQA and SF-Base-VQA datasets.

| Method | $\delta$ | Pitts-VQA | | | | SF-Base-VQA | | | |
|---|---|---|---|---|---|---|---|---|---|
| | | $\bar{A}_d$ | $\bar{A}_s$ | $\bar{A}$ | $A_{ds}$ | $\bar{A}_d$ | $\bar{A}_s$ | $\bar{A}$ | $A_{ds}$ |
| Satellite w/o road | 0.3 | 91.33 | 85.51 | 88.36 | 64.05 | 90.85 | 84.57 | 87.32 | 65.33 |
| Satellite | 0.3 | 92.05 | 86.78 | 89.32 | 68.98 | 91.52 | 85.79 | 88.67 | 70.42 |
| Satellite w/o road | 0.5 | 91.09 | 85.17 | 88.06 | 64.63 | 90.88 | 84.32 | 87.41 | 65.93 |
| Satellite | 0.5 | **92.70** | **87.46** | **90.02** | **69.60** | **92.06** | **86.66** | **89.33** | **70.45** |

Table 3: Ablation study of training with different parameters during different training phrases, *i.e.* Vision Encoder (VE), VL Adapter (VLA), and LLM. ✔ indicates one module is trainable.

| Variants | Stage-1:Alignment Tuning | | | Stage-2:Localization Tuning | | | Pitts-VQA | | SF-Base-VQA | |
|---|---|---|---|---|---|---|---|---|---|---|
| | VE | VLA | LLM | VE | VLA | LLM | $\bar{A}$ | $A_{ds}$ | $\bar{A}$ | $A_{ds}$ |
| A | | ✔ | | | ✔ | ✔ | 86.58 | 63.21 | 86.74 | 62.94 |
| B | | ✔ | | ✔ | ✔ | ✔ | 86.42 | 62.95 | 86.31 | 62.78 |
| C | | ✔ | ✔ | | ✔ | ✔ | 87.48 | 63.03 | 85.92 | 61.21 |
| D | | ✔ | ✔ | ✔ | ✔ | ✔ | 89.53 | 66.37 | 89.63 | 68.95 |
| E | ✔ | ✔ | ✔ | | ✔ | ✔ | 87.37 | 63.52 | 87.07 | 64.68 |
| AddressVLM | ✔ | ✔ | ✔ | ✔ | ✔ | ✔ | 90.02 | 69.60 | 89.33 | 70.45 |

achieves the best results across all metrics on both datasets. Specifically, the zero-shot performance of LLaVA-Phi3-mini is subpar on both datasets, primarily due to its inadequate fine-grained and multi-modal understanding of urban environments. Nevertheless, it is worthy noting that its performance in district-level judgment ($A_d^J$) is better than random guessing (60.22% vs. 50% and 71.73% vs. 50% on both datasets), suggesting that it does have a foundational level of urban knowledge. After applying our two-stage tuning to LLaVA-Phi3-mini, there is a significant improvement in AddressVLM's overall performance compared to the zero-shot setting (+49.01% and +49.02% on both datasets in terms of $\bar{A}$), indicating that our framework can effectively enhance the model's image address localization capabilities. For the SOTA method GeoReasoner, the key lies in the first-stage reasoning tuning that aims at coarse-grained recognition and enhanced reasoning ability. While this strategy yields benefits at the country level, it has been observed that the limited distinctions in street scenes within the same city can lead to a detrimental effect, resulting in decreases of 2.74% and 2.63% in terms of $A_{ds}$ on both datasets. In contrast, our AddressVLM constructs a satellite image and street-view image alignment task in the first-stage tuning, effectively integrating knowledge about street names and global street distribution into the model. Compared to the baseline of directly applying localization tuning, the proposed alignment tuning stage brings significant and consistent performance gains, *e.g.*, +9.08% and +11.83% in terms of $A_{ds}$ on both datasets. Furthermore, we can observe a performance gap between our AddressVLM and AddressCLIP in terms of street and district localization performance ($A_{ds}$), suggesting that it is still challenging for open-set generative models to achieve comparable results as closed-set classification models in specific tasks. This is a promising direction and we would like to explore it in future work.

## 4.3 ABLATION STUDY

**Grafting Mechanism of Cross-view Alignment Tuning.** The cross-view alignment tuning is a pivotal step for the effectiveness of AddressVLM, with various options for constructing the visual data. The first key factor is the overlap ratio $\delta$ (default $\delta = 0.5$) of the longer side of the street-view image to the satellite image. The second factor is the type of satellite images, *i.e.*, whether the satellite image is labeled with textual street names. The ablation results with different grafting ratios and satellite map types are shown in Tab. 2. It is shown that reducing $\delta$ to 0.3 leads to a decline in performance, indicating that excessively small street view images fail to provide sufficient visual details. Meanwhile, removing street labels from satellite images also results in performance degradation since satellite maps inadequately represent street layouts, which lack the OCR road information. Therefore, we finally adopt satellite images with street names and set $\delta = 0.5$.

**Training Components in LVLM.** Whether the training parameters in LVLMs are frozen or not usually affects its performance on domain-specific tasks. To this end, we explore the impact of freezing or unfreezing components of AddressVLM as shown in Tab. 3, which includes the Vision Encoder (VE), VL Adapter (VLA), and LLM. Our baseline setup (Variant A) involves unfreezing

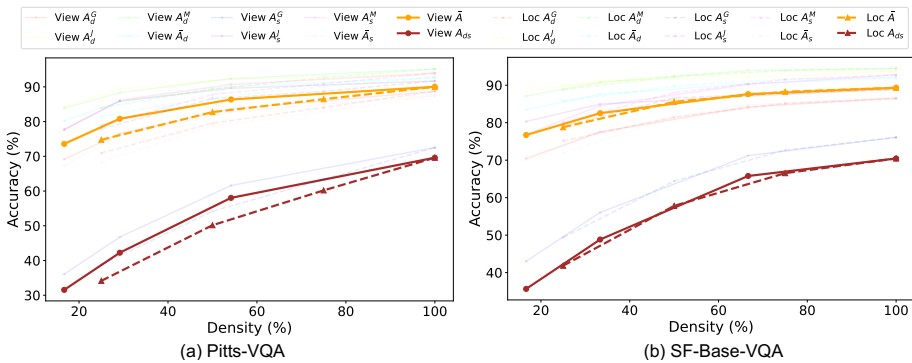

Figure 5: Ablation on different densities of street-view images for address localization on the Pitts-IAL and SF-IAL-Base datasets.

Table 4: Effect of mixed training on both Pitts-IAL and SF-IAL-Base datasets.

| Train / Test | District | | | | Street | | | | $\bar{A}$ | $A_{ds}$ |
|---|---|---|---|---|---|---|---|---|---|---|
| | $A_d^G$ | $A_d^J$ | $A_d^M$ | $\bar{A}_d$ | $A_d^G$ | $A_d^J$ | $A_d^M$ | $\bar{A}_d$ | | |
| Pitts / Pitts | 88.73 | 93.54 | 95.16 | 92.70 | 72.51 | 91.70 | 93.98 | 87.46 | 90.02 | 69.60 |
| Pitts + SF / Pitts | 89.24 | 93.17 | 95.16 | 92.66 | 72.90 | 92.77 | 94.34 | 88.18 | 90.37 | 70.63 |
| SF / SF | 86.48 | 93.72 | 94.50 | 92.06 | 76.09 | 88.92 | 92.75 | 86.66 | 89.33 | 70.45 |
| Pitts + SF / SF | 87.40 | 94.24 | 94.92 | 92.66 | 77.05 | 91.97 | 93.00 | 88.48 | 90.55 | 71.36 |

the VLA during both two stages, while the LLM is unfrozen solely in the second stage. Notably, unfreezing the LLM during the first stage yields the most substantial performance improvement. Similarly, unfreezing the VE in the second stage usually achieves better performance than freezing the VE, since the target of the second stage training is street-view images and unfreezing the VE enables the model to better adapt to urban street scenes. Ultimately, unfreezing all parameters leads to the best performance. This result can be attributed to the task's strong specificity and the availability of a large-scale dataset, which facilitates comprehensive parameter optimization for optimal results. These findings align with previous conclusions in the community (Lin et al., 2024).

**Density of Street-view Images.** We investigate the impact of different densities of street-view images used for the address localization tuning, which can be reflected in two aspects: i) The density of viewpoints, meaning how many street views are available for a single location (*e.g.*, 100%, 50%, 25%, 12.5%). ii) The density of locations, referring to the down-sampling rate of locations (*e.g.*, 100%, 75%, 50%, 25%). We decouple these factors for separate analysis as shown in Fig. 5. As observed, in terms of viewpoint density, the model maintains over 88% performance when the number of street views exceeds 6 in terms of ($\bar{A}$). For location density, the model retains over 71% performance even when locations are down-sampled to 50% in terms of ($A_{ds}$). The results indicate that our approach has strong generalization capabilities even with lower data densities. Meanwhile, we notice that the sensitivity of our method to viewpoint and location density is similar, which suggests that the density of these two dimensions is equally significance to the localization performance.

**Scalability for Multiple Cities.** Considering that image address localization may involve multiple cities in practice, we evaluate the scalability of AddressVLM on the Pitts-VQA and SF-Base-VQA datasets. Specifically, we merge these datasets and train a unified AddressVLM using the proposed two-stage tuning, then evaluate it on both test sets. As shown in Tab. 4, surprisingly, the performance of this unified model surpasses the performance of each separate model slightly on both datasets. We speculate that more cross-view data of the same task facilitates model learning how to locate the street-view image using a map for reference. This finding further demonstrates the scalability of our pipeline, suggesting its potential to extend capabilities across more cities or even an entire country.

### 4.4 QUALITATIVE RESULTS

**Effectiveness of Cross-view Alignment Tuning.** To demonstrate the effectiveness of the proposed cross-view alignment tuning on the final address localization quality, we present examples with correct positioning by our model with the alignment tuning, as shown in Fig. 6. We present street-view

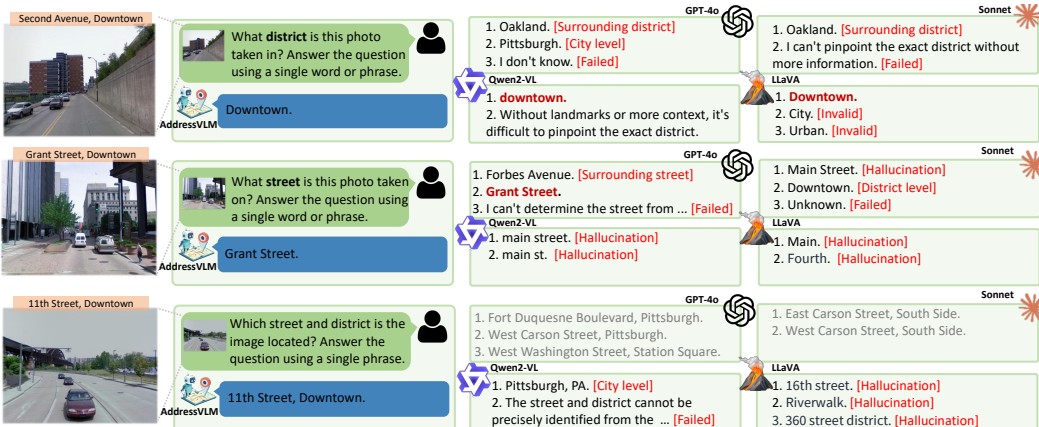

Figure 6: Qualitative visualization comparison of the impact of whether using the first-stage cross-view alignment tuning. The street-view images around the mispredicted streets are also depicted.

Figure 7: Qualitative comparison of address question-answering capabilities with general LVLMs.

images that are predicted incorrectly without the first-stage alignment tuning. It can be observed that there exists high degree of similarity between the street views near the mispredicted streets and those of the ground truth streets. This challenge is difficult to address by only using the second-stage address localization tuning. In contrast, the first-stage alignment tuning supplements the missing global street information and establishes connections between street-view images, thus helping the model better confirm the location of the street-view image during the address localization stage.

**Comparisons with General LVLMs.** We further present examples of AddressVLM in real-world inference and provide a qualitative comparison with SOTA general LVLMs, *e.g.*, GPT-4o (Achiam et al., 2023), Sonnet 3.5 (Claude, 2024), and Qwen2-VL (Qwen, 2024; Bai et al., 2023) and LLaVA-Phi3-mini, as shown in Fig. 7. Our approach consistently delivers high-quality results across various VQA scenarios. In contrast, the performance of SOTA models is significantly constrained by whether the input images contain sufficient identifiable information, such as street names and landmarks. This demonstrates that with minimal fine-tuning, AddressVLM can achieve a granular understanding of urban environments using only 4B parameters. This ensures its feasibility for future on-device deployment and updates.

## 5 CONCLUSION

In this work, we propose AddressVLM for city-wide address localization, which can perform flexible address question-answering for street-view images. The core idea is to leverage cross-view alignment tuning between satellite-view images and street-view images to integrate a global understanding of street distribution into LVLM. This contains two key components, namely the satellite and street view image grafting mechanism, and the automatic alignment label generation mechanism. The model undergoes two-stage fine-tuning, including cross-view alignment tuning and address localization tuning. Extensive experiments show that the proposed AddressVLM surpasses general LVLMs and SOTA localization LVLMs, and can be extended to multiple cities. In future work, we would like to explore cities on different continents and adopt larger LVLMs.

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

APPENDIX

# A    IMPLEMENTATION DETAILS

All our experiments are conducted using the xtuner framework on 8 RTX 3090 GPUs. The torch version is 2.4.0, the CUDA version is 12.1, and the transformers version is 4.37.2. The main hyper-parameter settings are given in Tab. 5.

Table 5: Hyper-parameter settings of the both two tuning stage.

| Hyper-parameter | Values |
|---|---|
| Batch Size | $4 \times 8$ |
| Gradient Accumulation | 16 |
| Learning Rate | 1e-5 |
| Weight Decay | 0 |
| Betas | (0.9, 0.999) |
| Warmup Ratio | 0.03 |
| LoRA Rank | 128 |
| LoRA Dropout | 0.05 |
| Model Max Length | $2048 - (336/14)^2$ |

# B    DATASETS DETAILS

We provide detailed information about the two constructed VQA datasets as a supplementary to Sec. 3.3, listed in Tab. 6. The dataset information includes the number of locations, the number of street view images, and the proportions of various dialogue types in the muti-turn conversations for both Pitts-VQA and SF-Base-VQA datasets. Generally, the distribution of address question types in the training set is balanced (1:1:1). In the test set, to accommodate both answer types (Yes/No) in judgment questions, we increased the judgment questions for each district-related and street-related question with answers set as "Yes" or "No", respectively. As a result, the proportion of judgment questions is nearly twice that of the generation and multiple-choice questions.

Table 6: More details of the constructed Pitts-VQA and SF-VQA datasets.

| Statistics | Pitts-VQA | | SF-Base-VQA | |
|---|---|---|---|---|
| | Train | Test | Train | Test |
| Covered Area | $20 \text{ km}^2$ | $20 \text{ km}^2$ | $6 \text{ km}^2$ | $6 \text{ km}^2$ |
| Number of locations | 7410 | 798 | 11946 | 1707 |
| Number of Districts | 19 | 19 | 15 | 15 |
| Number of Streets | 194 | 165 | 121 | 110 |
| Number of images | 177840 | 19152 | 143352 | 20484 |
| Number of questions | 533520 | 168409 | 430056 | 181943 |

Additionally, the question templates for different types of questions and address is given in Tab. 7. Each address type includes 10 distinct templates, resulting in 20 templates in total. Subsequently, different question types are generated by appending different prompts for the three question categories, as shown in Tab. 8. We replace the contents in "[]" with the ground truth location names (*e.g. street and district*) before appending them to the address prompts.

# C    VISUAL DATA CONSTRUCTION OF CROSS-VIEW ALIGNMENT TUNING

Multiple methods are available for constructing input images for cross-view alignment tuning, as illustrated in Fig. 8. The first method involves stitching the map and street view images at approximately a 1:1 ratio. This approach appears to preserve the most information from both the map and

Table 7: Question Templates for VQA Data Generation.

| Address Type | Template |
|---|---|
| District | Tell me the district where this image was captured. 
 I'm curious about the district, where is this? 
 In which urban district was this photo taken? 
 Can you identify which district this is? 
 What district is shown in this photograph? 
 What major district does the photo fall under? 
 I'm looking for the name of the district in this photo, can you help? 
 Can you specify the district shown in this photo? 
 Which district is depicted in the photo? 
 What's the name of the district shown in the photo? |
| Street | Identify the street in this image, please. 
 What is the street seen in this picture called? 
 On which boulevard or street was this taken? 
 Give me the name of the street that appears in this photograph. 
 Where was this, can you name the street? 
 What's the name of the avenue or street captured in this shot? 
 The street in this image, what is it named? 
 What's the name of this street shown in the photo? 
 Can you tell me which road this is? 
 What thoroughfare is depicted here? |

Table 8: Appended Prompts to Generate Different Question Types.

| Question Type | Template |
|---|---|
| Generation | Answer the question using a single word or phrase. |
| Judgement | Is this image taken [On STREET/IN DISTRICT], Yes or No? |
| Multiple Choice | Which of the following [STREET/DISTRICT] correctly represents the location shown in the image? 
 (A) [OPTION A] (B) [OPTION B] (C) [OPTION C] (D) [OPTION D]. 
 Please select the correct option (A/B/C/D). |

street view. However, since most LVLMs only accept square-shaped input images (*e.g.*, 336×336), the necessary padding and resizing operations result in a decreased number of effective visual tokens, which is detrimental to model learning. The second method entails inputting the two images separately. While this strategy allows for maintaining distinct features of both images, it may lead the model to overly rely on the street view content at the expense of the map information. Additionally, this approach effectively doubles the number of visual tokens, negatively impacting training efficiency. To mitigate these issues and encourage the model to focus on the overall street distribution information from the map, while also conforming to the LVLM input size requirements and ensuring training efficiency, we adopt the third method for visual data construction. The size of the map is resized to 336×336 to fit the input size of LVLMs.

## D    REPRODUCTION OF GEOREASONER

The training process for GeoReasoner (Li et al., 2024) consists of two stages. The first stage involves coarse-grained localization at the country level, accompanied by intricate reasoning derived from game data. The second stage is centered on fine-grained localization at the city level, utilizing Google Street View data. In our study, we replicate this pipeline to achieve district and street-level localization within the same urban area. A primary distinction between GeoReasoner and our AddressVLM lies in the data employed during the first stage. In the original work of GeoReasoner, the first stage data integrates external knowledge sourced from real geo-localization games. For

Figure 8: Three ways of combining the satellite-view and street-view images.

(a) Two images stitched    (b) Two images input separately    (c) Two images grafted

**GeoReasoner Prompt:**
Examine the image presented, and deduce from which neighborhood it was most likely captured, providing your rationale. (**Hint: This image was taken in North Beach within San Francisco**. Begin your response by confirming its location in North Beach, then substantiate your claim by analyzing the distinct architectural and streetscape style characteristic of North Beach that in this image.

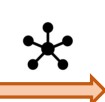

LLaVA-v1.6-Mistral-7B

**GeoReasoner Label:**
The image is indeed captured in North Beach, San Francisco. The distinct architectural and streetscape style characteristic of North Beach can be observed in the white building with a green awning. The building's design, along with the presence of a tree and a car parked on the street, suggests a quaint and charming neighborhood. The North Beach neighborhood is known for its Italian heritage and is characterized by narrow streets, historic buildings, and a vibrant cultural scene. The image captures the essence of this neighborhood, making it a likely location within North Beach.

Figure 9: An example of the prompt and the generated reasoning label for the first stage of GeoReasoner. The model of LLaVA-v1.6-Mistral-7B is adopted for label generation.

district-level localization, we generate reasoning data by emulating the reasoning generation pipeline utilized for our cross-view tuning data, as detailed in Appendix C. An example of the prompt and the generated reasoning label for the first stage of GeoReasoner is presented in Fig. 9. To facilitate a comprehensive comparison across various metrics outlined in Sec. 4.1, we employ the same VQA data for training the second stage of GeoReasoner.

# E  IMPLEMENTATION DETAILS OF QUALITATIVE RESULTS

## E.1  QUALITATIVE RESULTS IN SEC. 3.2

In Fig. 3, we conduct a quantitative analysis of the cross-view alignment tuning by examining the outputs from two distinct models. While the first stage utilizes grafted images as inputs, our principal objective is to establish a connection between street-view images and the street addresses. Consequently, we employ only street-view images as the input for this analytical evaluation.

**After Cross-view Alignment Tuning.** For discriminative models like CLIP, we can compare the embeddings of street views and address texts to assess whether the model effectively associates street layouts with street views. However, this method is not suitable for the generative models discussed in this study. Instead, we leverage the inherent randomness in the output of generative models. Specifically, we increase the temperature of the model during inference from 0.1 to 0.8 to encourage output variability. By performing inference for 100 times on the same input image, we can count the number of different valid streets, approximating the output distribution for the model for a given input.

**Before Cross-view Alignment Tuning.** Since the image address localization task is quite challenging, the model without any downstream fine-tuning (zero-shot model) struggles to produce valid street outputs directly. Therefore, we organize all the street names generated by the model above into options, allowing the zero-shot model to select one street from this given list for output. The difference between the prompts of these two models is given in Fig. 10.

## E.2  QUALITATIVE RESULTS IN SEC. 4.4

In Sec. 4.4, we demonstrate the results of four current state-of-the-art proprietary and open-source models on several samples in our datasets. Our AddressVLM is capable of generating outputs

Figure 10: Prompts for models before and after cross-view alignment tuning for qualitative results in Sec. 4.4.

Table 9: Detailed results of the ablation studies on the complementary metrics.

| | Ablations | District | | | | Street | | | | $\bar{A}$ | $A_{ds}$ |
|---|---|---|---|---|---|---|---|---|---|---|---|
| | | $A_d^G$ | $A_d^J$ | $A_d^M$ | $\bar{A}_d$ | $A_s^G$ | $A_s^J$ | $A_s^M$ | $\bar{A}_s$ | | |
| Pitts-VQA | Satellite w/o road (0.3) | 85.93 | 93.00 | 93.70 | 91.33 | 67.24 | 91.18 | 92.55 | 85.51 | 88.36 | 64.05 |
| | Satellite (0.3) | 87.32 | 92.93 | 94.59 | 92.05 | 71.61 | 91.04 | 93.27 | 86.78 | 89.32 | 68.98 |
| | Satellite w/o road (0.5) | 86.23 | 92.42 | 93.54 | 91.09 | 67.97 | 90.50 | 91.79 | 85.17 | 88.06 | 64.63 |
| | Variant A | 85.57 | 90.73 | 92.03 | 89.77 | 65.60 | 89.42 | 90.39 | 83.90 | 86.58 | 63.21 |
| | Variant B | 85.48 | 90.65 | 92.12 | 89.59 | 65.05 | 89.21 | 90.02 | 83.44 | 86.42 | 62.95 |
| | Variant C | 84.86 | 91.98 | 92.85 | 90.34 | 66.39 | 90.63 | 91.46 | 84.75 | 87.48 | 63.03 |
| | Variant D | 87.36 | 93.23 | 95.08 | 92.66 | 71.19 | 91.58 | 93.85 | 87.02 | 89.53 | 66.37 |
| | Variant E | 85.00 | 92.02 | 92.65 | 90.34 | 66.64 | 90.27 | 91.05 | 84.54 | 87.37 | 63.52 |
| | View-4/24 | 69.14 | 84.29 | 83.90 | 80.21 | 36.08 | 77.68 | 77.71 | 67.25 | 73.58 | 31.55 |
| | View-7/24 | 76.54 | 89.38 | 88.33 | 85.73 | 46.75 | 85.90 | 86.14 | 76.13 | 80.83 | 42.25 |
| | View-13/24 | 83.67 | 92.28 | 92.34 | 90.04 | 61.60 | 89.58 | 90.69 | 82.84 | 86.36 | 58.04 |
| | Location-1/4 | 70.95 | 85.91 | 83.97 | 81.47 | 38.81 | 77.91 | 78.92 | 68.35 | 74.76 | 34.14 |
| | Location-2/4 | 79.53 | 89.62 | 89.40 | 86.91 | 54.17 | 86.59 | 87.79 | 78.76 | 82.74 | 50.16 |
| | Location-3/4 | 84.06 | 92.18 | 92.69 | 90.18 | 63.34 | 88.73 | 90.88 | 82.90 | 86.46 | 60.19 |
| | AddressVLM | 88.73 | 93.54 | 95.16 | 92.70 | 72.51 | 91.70 | 93.98 | 87.46 | 90.02 | 69.60 |
| SF-Base-VQA | Satellite w/o road (0.3) | 84.11 | 91.82 | 92.64 | 90.85 | 73.59 | 88.38 | 90.51 | 84.57 | 87.32 | 65.33 |
| | Satellite (0.3) | 85.88 | 93.10 | 93.92 | 91.52 | 75.27 | 88.04 | 92.18 | 85.79 | 88.67 | 70.42 |
| | Satellite w/o road (0.5) | 84.39 | 91.85 | 92.79 | 90.88 | 73.87 | 88.35 | 90.68 | 84.32 | 87.41 | 65.93 |
| | Variant A | 82.92 | 92.60 | 92.41 | 90.07 | 69.29 | 88.27 | 88.13 | 83.47 | 86.74 | 62.94 |
| | Variant B | 82.90 | 92.50 | 92.03 | 89.92 | 68.90 | 87.14 | 87.97 | 82.77 | 86.31 | 62.78 |
| | Variant C | 82.05 | 91.99 | 92.26 | 89.51 | 67.90 | 87.13 | 87.53 | 82.39 | 85.92 | 61.21 |
| | Variant D | 85.87 | 94.73 | 95.16 | 92.57 | 74.60 | 90.22 | 92.05 | 86.76 | 89.63 | 68.95 |
| | Variant E | 83.55 | 92.25 | 92.75 | 90.15 | 71.28 | 87.91 | 89.20 | 84.06 | 87.07 | 64.68 |
| | View-2/12 | 70.47 | 88.39 | 87.11 | 83.47 | 43.01 | 80.31 | 76.87 | 70.08 | 76.71 | 35.65 |
| | View-4/12 | 77.52 | 91.18 | 90.74 | 87.57 | 56.05 | 84.97 | 84.54 | 77.59 | 82.53 | 48.83 |
| | View-8/12 | 84.23 | 92.11 | 93.94 | 90.34 | 71.20 | 87.12 | 90.23 | 84.74 | 87.63 | 64.72 |
| | Location-1/4 | 75.14 | 89.77 | 88.84 | 85.78 | 49.39 | 79.06 | 80.04 | 71.95 | 78.80 | 41.83 |
| | Location-2/4 | 81.52 | 92.69 | 92.27 | 89.72 | 64.42 | 86.62 | 88.02 | 81.49 | 85.56 | 57.81 |
| | Location-3/4 | 85.17 | 93.85 | 93.92 | 91.64 | 72.70 | 87.80 | 91.55 | 84.94 | 88.26 | 66.53 |
| | AddressVLM | 86.48 | 93.72 | 94.50 | 92.06 | 76.09 | 88.92 | 92.75 | 86.66 | 89.33 | 70.45 |

as the requirement in the prompt. However, the outputs of other LVLMs are more diverse and uncontrollable. Therefore, for each sample, we conduct multiple inferences (5-10 times) for each input, and display several most frequently responses.

## F  DETAILED RESULTS OF ABLATION STUDIES

We provide the detailed results of the ablation studies under all the metrics in Tab 9.

## G  MORE QUALITATIVE RESULTS

**Case Study.** We demonstrate more examples where AddressVLM accurately locates while the baseline model without cross-view alignment tuning makes errors in localization, as shown in Fig. 11. We also provide some failure cases that both model can not localize correctly in Fig. 11. One can see that these images are of low visual cues, which are difficult to recognize even for human experts.

**More Comparisons with General LVLMs.** Fig. 12 demonstrates more qualitative results and comparisons between various general LVLMs given different types of input prompts and images.

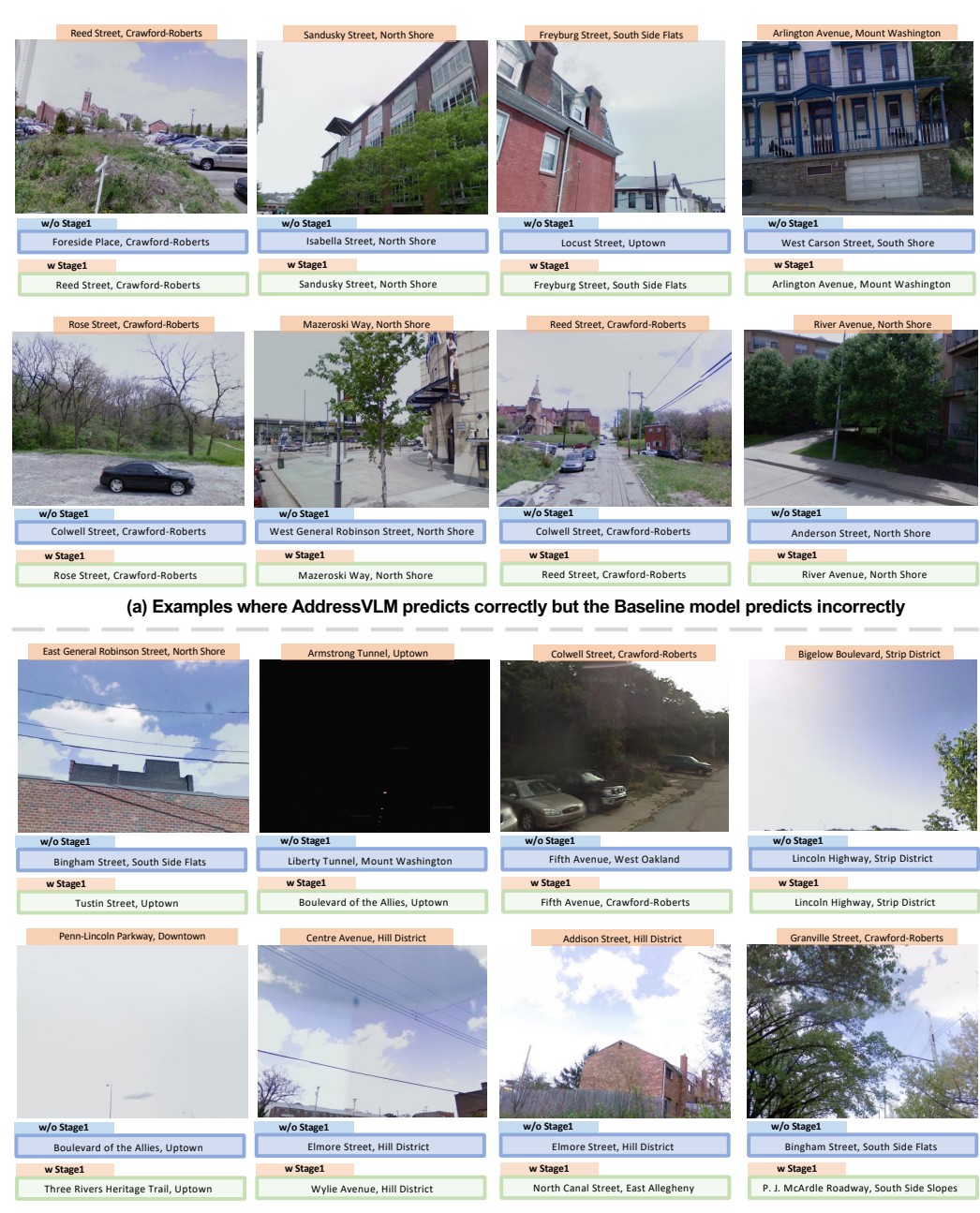

(a) Examples where AddressVLM predicts correctly but the Baseline model predicts incorrectly

(b) Failure Cases

Figure 11: More examples where AddressVLM accurately locates while the baseline model makes errors in localization (a), as well as failure cases (b).

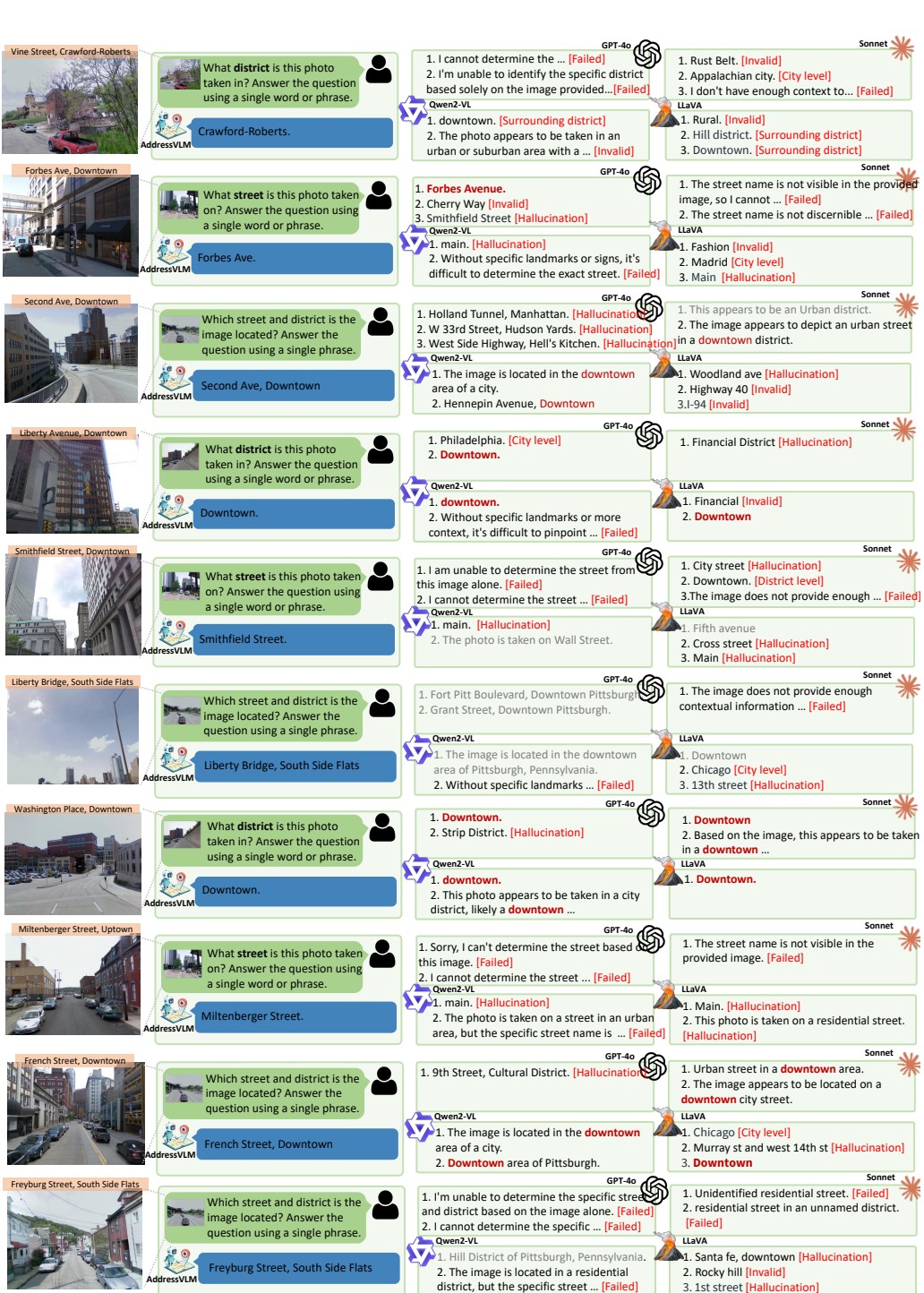

Figure 12: More qualitative examples of comparison with the general LVLMs.

