# OpenReview forum: "AddressVLM: Cross-view Alignment Tuning for Image Address Localization using Large Vision-Language Models"
_ICLR.cc/2025/Conference — Submitted to ICLR 2025_

### Official Review · Reviewer_pFw4 · 2024-10-30

**Soundness:** 2
**Presentation:** 3
**Contribution:** 3
**Rating:** 5
**Confidence:** 5

**Summary:**

This paper presents AddressVLM, a novel approach for enhancing fine-grained street-level localization in urban areas using LVLMs through the incorporation of satellite imagery and cross-view alignment. AddressVLM introduces a satellite-view and street-view image grafting mechanism, along with an automatic alignment label generation mechanism. This helps build connections between street-view images through cross-view matching, thereby enhancing the LVLM’s global understanding of street distributions. The model uses a two-stage training protocol: Cross-view alignment tuning, which establishes spatial correlations by combining cross-view images and generating labels automatically, followed by Address localization tuning to further optimize accuracy. The authors created two street-view VQA datasets, Pitts-VQA and SF-Base-VQA, based on image address localization datasets from Pittsburgh and San Francisco, providing valuable resources for evaluating fine-grained urban localization within the community.

The main contribution of this work is the cross-view alignment tuning method, which integrates macro and micro visual cues from urban environments into LVLMs. By grafting street-view and satellite images and employing automated label generation, AddressVLM enhances the model’s global understanding of street distributions, showing significant improvements compared to baseline LVLMs and the state-of-the-art GeoReasoner model. Experimental results indicate that AddressVLM improves accuracy on the Pitts-VQA and SF-Base-VQA datasets by 9% and 12%, respectively.

Overall, this paper demonstrates an innovative use of LVLMs for city-level localization; however, some clarifications on specific points in the theoretical and experimental sections are still needed.

**Strengths:**

1. The cross-view alignment tuning in AddressVLM addresses the gap in fine-grained street-level localization for LVLMs, a challenging problem that previous work has not fully resolved. This is achieved through a novel image grafting mechanism and automatic label generation, which enhances the model's ability to recognize urban street patterns.

2. The authors introduce two VQA datasets specifically tailored for image address localization. It is hoped that the datasets used for the two-stage training, as well as the trained models and weights, can be made publicly available.

3. The paper is clearly and well written.

**Weaknesses:**

1. The experimental results lack evidence of the replicated GeoReasoner’s performance at the city and country levels. Is it comparable to the results in the original paper? This makes it difficult to determine whether the replication of GeoReasoner is reasonable.

2. Is there a comparison with UrbanCLIP and UrbanVLP?

3. The paper concludes that the optimal overlap ratio δ between the longer side of the street-view image and the satellite image is 0.5, but more extensive ablation experiments on different overlap ratios, such as values greater than 0.5, are missing. Quantitative experiments on the three grafting methods are also needed.

4. There is a lack of quantitative experiments to demonstrate whether performing the first stage Cross-view alignment tuning has a significant impact.

**Questions:**

In addition to Weaknesses, I have some confusion：
1. What datasets were used in the first and second stages? Were any additional datasets included? In the second stage, Address localization tuning uses formatted data for fine-tuning—does this impact the model’s performance on other tasks or dialogue capabilities? Are there experiments evaluating this effect?

2. Can the final trained LVLM model only output brief district names as shown in the paper? Is the LVLM capable of providing reasons or analyses that explain how it arrived at the location conclusion?

3. Given that Address localization tuning relies on the formatted VQA dataset proposed in the paper, would the model still return accurate district locations if questions were asked in a free-form style rather than constrained by templates?

4. In the Automatic Alignment Label Generation step, was another LVLM used, such as GPT-4?

5. In section 4.3, the paper states that unfreezing the Vision Encoder (VE) during the second stage generally improves performance compared to keeping it frozen. While this is apparent when comparing CD with (E, AddressVLM), it’s not clear when comparing AB. Why is this?

6. Does the street-view image need to be grafted onto the upper right corner of the satellite view? Would other positions work as well?

7. How does the model’s performance compare to other traditional geo-localization methods, such as Pigeon or GeoCLIP?

---

> ### Author Response · Authors · 2024-11-20
>
> Thank you very much for the constructive feedback! Below, we address the concerns raised in the review and will adjust the paper accordingly in the revision.
>
> **Response to weakness 1**: First, the reason analysis introduced by GeoReasoner in the first tuning stage is designed for coarse-grained country and city-level address localization, and no global information is introduced when expanded to fine-grained street-level address localization. In contrast, our method introduces satellite images as global information in the first tuning stage, integrating the global understanding of urban street distribution into LVLMs. This is the key factor why our method is superior to GeoReasoner. Second, due to the lack of access to GeoReasoner's data and the corresponding benchmark data for its second phase, we are unable to reproduce its results on the benchmark. We fully followed the method description in the original text and specifically constructed reason labels in the first stage for fine-tuning for GeoReasoner to reproduce the method to the greatest extent possible. Third, should GeoReasoner releases its code or data in the future, we will conduct the necessary evaluation or update our results accordingly.
>
> **Response to weakness 2**: Since UrbanCLIP and UrbanVLP are designed to solve the urban indicator prediction problem, rather than our image address localization problem, these two methods are difficult to directly extend to the address localization task. Therefore, we cannot compare with UrbanCLIP and UrbanVLP: 1) In terms of input, these two methods typically require satellite images as input, while our method only requires street view images for inference. 2) In terms of output, these two methods focus on predicting numerical indicators such as Carbon, Population, and GDP in downstream tasks, while our method focuses on address related VQA.
>
> **Response to weakness 3**: Thanks for your suggestion! The table below presents the results on Pitts-VQA when $\delta$ is set to 0.3, 0.5, and 0.7, as well as a comparison of the outcomes for three different grafting methods. It can be observed that the results are superior when $\delta=0.5$, it outperforms both $\delta=0.3$ and $\delta=0.7$. This is because when $\delta$ is too small, there is insufficient fine-grained information from the street view images, and when delta is too large, it obscures most of the map area, leading to a lack of information from the map. Comparing the three different grafting methods, our grafting approach yields the best results. This is attributed to the fact that LLaVA-phi3-mini is designed for single-image input scenarios, creating a significant domain gap when two images are inputted (**two images stitched**), deviating from its pre-training conditions. Additionally, since CLIP is adapted to square input images, images with a high aspect ratio may lose token information due to resizing and padding operations (**two images stitched**). In summary, we have chosen the $\delta$ of 0.5 with **two images grafted** for input image construction.
>
> | Method | $\delta$ | $A_d^G$ |  $A_d^J$ |  $A_d^M$ |  $\bar{A}_d$ | $A_s^G$ |  $A_s^J$ |  $A_s^M$ |  $\bar{A}_s$ | $\bar{A}$ | $A_{sd}$ |
> | --- | --- | --- | --- | --- | --- | --- | --- | --- | --- | --- | --- |
> | Satellite | 0.3 | 87.32 | 92.93 | 94.59 | 92.05 | 71.61 | 91.04 | 93.27 | 86.78 | 89.32 | 68.98 |
> | Satellite | 0.5 | 88.73 | 93.54 | 95.16 | 92.70 | 72.51 | 91.70 | 93.98 | 87.46 | 90.02 | 69.60 |
> | Satellite | 0.7 | 86.58 | 92.33 | 93.86 | 91.22 | 70.35 | 90.49 | 92.65 | 85.81 | 88.49 | 68.07 |
>
> | Method | $A_d^G$ |  $A_d^J$ |  $A_d^M$ |  $\bar{A}_d$ | $A_s^G$ |  $A_s^J$ |  $A_s^M$ |  $\bar{A}_s$ | $\bar{A}$ | $A_{sd}$ |
> | --- | --- | --- | --- | --- | --- | --- | --- | --- | --- | --- |
> | Two image stitched | 85.73 | 92.39 | 93.42 | 90.95 | 67.36 | 91.05 | 92.17 | 85.78 | 87.91 | 66.45 |
> | Two images input separately | 86.52 | 92.90 | 94.15 | 91.33 | 68.29 | 90.84 | 92.71 | 86.67 | 88.14 | 67.39 |
> | Two images grafted | 88.73 | 93.54 | 95.16 | 92.70 | 72.51 | 91.70 | 93.98 | 87.46 | 90.02 | 69.60 |
>
> **Response to weakness 4**: In Table 1, the **baseline** represents the results of directly fine-tuning on VQA data (stage2) without undergoing stage 1 training. It can be observed that our cross-view alignment tuning consistently and significantly enhances performance across all metrics (e.g., +8.2% on $A_s^G$, +9.08% on $A_{sd}$). Please kindly check it.

---

> > ### Author Response · Authors · 2024-11-20
> >
> > **Response to question 1**: 1) The training data for the first and second phases are geographically consistent, but they differ in the form of input images and QA content. The first phase uses satellite images and street views in terms of vision, and a single-round conversation for cross-view alignment in terms of text. The second phase uses only street-views and multiple rounds of conversations related to addresses. 2) No additional data is used throughout the entire process. 3) The formatted data in the second phase may have a negative impact on the model's general VQA capabilities. Nevertheless, the impact can be mitigated by incorporating general VQA data into the second phase of training. For instance, we can use the 515K VQA data from LLaVA.
> >
> > **Response to question 2**: The generation of brief answers is also included in our training and testing process. Thanks to the cross-view alignment phase of training, our model is capable of providing reasons for its localization decisions, although we have not designed specific metrics to test it.
> >
> > **Response to question 3**: Regardless of the form in which users pose questions, our model can still provide accurate answers. This is because we utilized a variety of question and answer templates when constructing our VQA data, preventing the model from overfitting to a specific questioning format. For more details, please refer to Table. 7 and Table. 8 in the appendix.
> >
> > **Response to question 4**: Thank you for the insightful discussion. There are no specific restrictions on the LVLM used in the Automatic Alignment Label Generation step, as long as it can provide reasons for cross-view alignment. We believe that employing the advanced GPT-4o as the LVLM for this step can result in more accurate labels. However, due to the potential high cost of annotation, scaling up to larger datasets may be challenging. Therefore, we use open-source models for annotation to achieve a balance between cost and effectiveness.
> >
> > **Response to question 5**: In variants AB, only the Vision-Language Adapter (VLA) was unfrozen during the first phase of training. This led to a strong dependence on modifications to the VLA for fitting the data, a process that could potentially disrupt the well-aligned VLA parameters from the pre-training. Consequently, in the second phase, when the vision encoder (VE) was unfrozen for training, the misaligned VLA and VE for co-optimization could prove challenging, which might not yield immediate benefits.
> > In contrast, variants CD unfrozen both the VLA and the LLM during the first phase. This allowed the LLM and VLA to share the task of fitting the data. As a result, the VLA remained well-aligned with the original VE, making it easier to optimize when the VE was unfrozen in the second phase for joint fine-tuning with the VLA and LLM.
> >
> > **Response to question 6**: No, the location of the street view image can be at any corner of the satellite image, as long as it does not obscure their true location on the map. Subsequently, the prompts for cross-view tuning can be modified accordingly. We choose the top-right corner simply because it aligns to some extent with our visual preferences.
> >
> > **Response to question 7**: As we mentioned in related work, both Pigeon and GeoCLIP focus on global country and city-level address localization, and the output is in the form of GPS coordinates. In contrast, our method focuses on fine-grained street-level address localization within the city, and is in the form of natural language question answering. Therefore, our method and these two methods are complementary and difficult to compare directly. Our method and these two methods can together form a complete system. For instance, for a given street view image, Pigeon or GeoCLIP is used to locate the country and city first, and then our method can be invoked to utilize the corresponding city model for fine-grained, semantically enriched address query and response.

---

> > > ### Comment · Reviewer_pFw4 · 2024-11-20
> > > **There are still a few questions.**
> > >
> > > Thanks.
> > >
> > > 1.**Reply to question 1:** I noticed the illustration in Figure 4 and understand that your design is divided into two stages: the first stage focuses on single-turn dialogues, while the second stage handles multi-turn dialogues. However, I observed that the two datasets you mentioned, Pitts-VQA and SF-Base-VQA, both contain 7 rounds of QA according to your description. This raises a question for me: is the first-turn dialogue data from a new dataset? For example, as shown in Figure 9, you concatenate satellite and street-view images, then use the LLM to generate reasoning, forming image-text pairs consisting of reasoning and concatenated images. Did you forget to mention this in Section 3.3?
> > >
> > > Additionally, in the second stage, are Pitts-VQA and SF-Base-VQA used simultaneously? If general datasets are not included, does this model only answer questions in a specific paradigm, with the questions needing to follow the format of your templates? If general data such as the 515K VQA dataset from LLaVA is included for fine-tuning, will the model’s performance in geolocation tasks degrade?I look forward to seeing an experimental result.
> > >
> > > 2.**Reply to question 2:** Regarding Question 2, it is well known that large language models (LLMs) can exhibit hallucination when generating reasoning or textual descriptions. Coupled with the scale issue of satellite imagery, some important features might also be overlooked. I am curious how you addressed these hallucination issues during the cross-view alignment training phase?
> > >
> > > 3.**Reply to question 3:** I know your Tables 7 and 8, what if I ask questions that are neither in Table 7 nor Table 8?
> > >
> > > 4.**Reply to question 4:** Is the LLM you used for the cross view justification description LLaVA-v1.6-Mistral-7B?
> > >
> > > 5.**Reply to question 7:** How does your model perform on other open-source benchmark datasets, such as Im2gps, Im2gps3k, YFCC4k, and OSV-5M? Specifically, how does it perform at 1km, 25km, 200km, 750km, and 2500km accuracy levels on these datasets? Additionally, what is your GeoScore? This is a crucial metric for evaluating performance in geolocation tasks.
> > >
> > > Looking forward to your reply.

---

> > > > ### Author Response · Authors · 2024-11-21
> > > >
> > > > Thank you for your prompt feedback and further discussion. Below, we address the further concerns and will adjust the paper accordingly in the revision.
> > > >
> > > > **Response to question 1**: Thank you for the feedback. We apologize for not fully understanding the reviewer in the previous round of responses and replying that no additional datasets were added, when we meant that no datasets other than street view and satellite images were added. We acknowledge that the datasets used in Stage 1 for Pittsburgh and San Francisco are different from Pitts-VQA and SF-Base-VQA. The Stage 1 datasets include satellite and street view images for the visual component and consist of only a single round of dialogue for the text component, making them essentially new datasets. While we did not specifically mention this in Section 3.3, as these datasets are used solely for training rather than for metrics evaluation, we plan to release the Stage 1 training datasets for both cities to the community. We will also include a description of this in the revised version.
> > > >
> > > > In the main experiments, we conducted the first and second stages of fine-tuning and evaluation separately on Pitts-VQA and SF-Base-VQA datasets. Additionally, we experimented with combining both datasets for simultaneous training. The results of this mixed training are presented in Table 4.
> > > >
> > > > AddressVLM can effectively respond to a wide range of questions related to addresses, even if these questions are not explicitly listed in Table 7 and Table 8. This is because the well-trained LVLM already has good instruction following and generalization abilities. Regarding the experiment you referred to, while the datasets used in the first stage remained unchanged, we included the llava_v1_5_mix665k dataset (https://huggingface.co/datasets/liuhaotian/LLaVA-Instruct-150K/blob/main/llava_v1_5_mix665k.json ) in the second stage for fine-tuning address localization. We anticipate that the inclusion of the general VQA dataset will have a limited impact on address localization capabilities. We expect the experiment to be completed within this week, and the results will be shared promptly once available.
> > > >
> > > > **Response to question 2**: We agree that LLMs typically suffer from hallucination problems. However, we can ensure the factual accuracy of cross-view alignment labels generated in Stage 1 depending on three aspects. 1) The label generation task in Stage 1 essentially belongs to the image caption task, which is one of the fundamental tasks that modern LVLMs like LLaVA-v1.6-Mistral-7B excel at. 2) As mentioned in lines 261-265 of our paper, the precise location of the street-view image is provided to the pre-trained LVLM in advance (hint: {near Forbes Avenue, Downtown, Pittsburgh}). This ensures that the model is explaining the reasons for cross-view alignment rather than predicting the reasons, thus avoiding potential errors or biases. 3) We manually check the image descriptions and alignment reasons to ensure that they are correct.
> > > >
> > > > **Response to question 3**: Sorry for our misunderstanding of your original concren. When asking questions that are not in Table 7 and Table 8, our model is still able to answer as long as they are semantically related to the address. This is because the well-trained LVLM already has good instruction following and generalization abilities, and the question templates in Table 7 and Table 8 can be seen as few samples that teach the model to answer address related questions.
> > > >
> > > > **Response to question 4**: Yes. We adopt LLaVA-v1.6-Mistral-7B as the LVLM for the alignment label generation.
> > > >
> > > > **Response to question 7**:  We appreciate the reviewer for providing the benchmarks and metrics for a comprehensive presentation. However, the benchmark and metrics you listed are aimed at planet-level geo-localization tasks, which predict the GPS coordinates of the country or city where the given image was token, thus enabling the evaluation metric at a distance scale of 750km or even 2500km. On the contrary, our focus is fine-grained address localization of urban streets and district within cities, where the scope of an entire city is typically within 25km. Therefore, these benchmarks and metrics do not match our research problem.
> > > >
> > > > On the other hand, calculating metrics at a scale like 1km requires a single GPS for each query image, but the street-level textual address estimated by AddressVLM corresponds to multiple GPS, making it impractical to evaluate the localization performance. That’s why we introduce the metrics  $\bar{A}_d$ and $\bar{A}_s$ that are calculated through text consistency.

---

> > > > > ### Author Response · Authors · 2024-11-25
> > > > > **Follow-up response to question 1**
> > > > >
> > > > > **Follow-up Response to question 1**
> > > > > The following are the experimental results after integrating the llava_v1_5_mix665k VQA dataset into the stage 2 training. It appears that our address localization performance has been somewhat affected, due to the smaller volume of address localization VQA data compared to the llava_v1_5_mix665k dataset (170k vs. 665k). We believe that by carefully adjusting the data ratio, we can strike a balance between enhancing the model's general abilities and its address localization capabilities.
> > > > >
> > > > > | Method | $A_d^G$ |  $A_d^J$ |  $A_d^M$ |  $\bar{A}_d$ | $A_s^G$ |  $A_s^J$ |  $A_s^M$ |  $\bar{A}_s$ | $\bar{A}$ | $A_{sd}$ |
> > > > > | --- | --- | --- | --- | --- | --- | --- | --- | --- | --- | --- |
> > > > > | AddressVLM | 88.73 | 93.54 | 95.16 | 92.70 | 72.51 | 91.70 | 93.98 | 87.46 | 90.02 | 69.60 |
> > > > > | w / llava_v1_5_mix665k | 74.92 | 88.33 | 86.72 | 84.38 | 46.07 | 86.25 | 84.40 | 75.70 | 79.95 | 42.08 |

---

> > > > ### Author Response · Authors · 2024-11-29
> > > > **Looking forward to your reply**
> > > >
> > > > We are very grateful for your feedback and our discussions have been incredibly insightful. We have conducted further experiments and provided responses to your questions. We are eager to hear if our responses have alleviated your concerns. We look forward to your reply and hope you enjoy your holiday!

---

> ### Author Response · Authors · 2024-12-02
> **Reviewer K9em commented in other Reviewer's block**
>
> To Reviewer **K9em**:
>
> We have to remind you that you have written your comments in the reply box belonging to Reviewer **pFw4**. The response to your further questions has been provided in the block belonging to yourself. Please kindly check it.

---

> ### Author Response · Authors · 2024-12-02
> **Follow-up response to question 1**
>
> To validate our assumption:
> **We believe that by carefully adjusting the data ratio, we can strike a balance between enhancing the model's general abilities and its address localization capabilities.**
> We conducted experiments by mixing address localization VQA data with llava_v1_5_mix665k data in a 1:1 ratio, and the results are shown in the table below. It can be observed that by increasing the proportion of address localization VQA data, the performance can be further enhanced.
> | Method | $A_d^G$ |  $A_d^J$ |  $A_d^M$ |  $\bar{A}_d$ | $A_s^G$ |  $A_s^J$ |  $A_s^M$ |  $\bar{A}_s$ | $\bar{A}$ | $A_{sd}$ |
> | --- | --- | --- | --- | --- | --- | --- | --- | --- | --- | --- |
> | w / llava_v1_5_mix665k | 74.92 | 88.33 | 86.72 | 84.38 | 46.07 | 86.25 | 84.40 | 75.70 | 79.95 | 42.08 |
> | w / llava_v1_5_mix665k (1:1) | 80.38 | 90.78 | 90.47 | 87.97 | 54.68 | 87.02 | 87.70 | 79.08 | 83.43 | 50.53 |
>
> Please kindly check the responses.
>
> Looking forward to your reply！

---

### Official Review · Reviewer_hWQs · 2024-11-02

**Soundness:** 2
**Presentation:** 3
**Contribution:** 2
**Rating:** 5
**Confidence:** 4

**Summary:**

This paper proposes a method to integrate the address localization capability within a city into large-scale visual-linguistic models (LVLM), in order to achieve flexible address-based question answering based on street view images. The main contributions include exploring the integration of address localization capability within a city into LVLM, proposing a cross-view alignment fine-tuning method, and introducing the AddressVLM model. The experimental results show that AddressVLM significantly outperforms the baseline LVLM and the state-of-the-art GeoReasoner model on the street view VQA dataset and demonstrates the address localization capability across multiple cities.

**Strengths:**

1. It adopts a cross-view alignment fine-tuning strategy by aligning the sparsely collected street view images with globally consistent satellite images, which enhances the LVLM's understanding of the overall city street distribution. This helps address the challenges that cannot be solved by only using second-stage address localization fine-tuning.

2. Compared to general LVLM, this method can achieve fine-grained understanding of the urban environment using only 4B parameters, providing feasibility for future device deployment and updates. This highlights the practicality and efficiency of the method.

**Weaknesses:**

1. Although an innovative method for cross-view alignment of street-view images and satellite images was proposed, there is a lack of theoretical analysis and mathematical derivation of this method, which makes it difficult to deeply understand its principles and
limitations.

2. The experimental part is only evaluated in a limited urban area, which cannot fully verify the applicability and scalability of this method in a wider urban environment.

**Questions:**

1. In the introduction, the author introduces AddressCLIP, a method based on image and text alignment, points out the related shortcomings, and introduces his own methods. However, the author does not indicate whether his predecessors have done any work on this question-and-answer-based approach. The author needs to make this clear.

2. Please briefly explain the difference between Visual Place Recognition and Cross-view
Geo-localization in a few sentences.

3. In section 3.2, the author mentions that "street-view images are scaled down and grafted onto satellite images like CutMix data augmentation". Could the author briefly explain why other mixup data augmentation methods are not used? Why is cutmix applied?

---

> ### Author Response · Authors · 2024-11-20
>
> Thank you very much for the constructive feedback! Below, we address the concerns raised in the review and will adjust the paper accordingly in the revision.
>
> **Response to weakness 1**: We clarify the motivation of the cross-view alignment fine-tuning and the limitations of the proposed method.
>
> * Motivation
>
> To realize the street-level address localization within a city, a reasonable way involves fine-tuning a well-trained LVLM using street-view question-and-answer (VQA) data with LoRA. However, this straightforward method yields suboptimal performance because street-view images are sparsely collected in terms of both location and viewpoint, which inhibits the model’s ability to build a global understanding of street distribution across the entire city. Such global information is crucial for effective address localization since street-view images are densely sampled during testing.
>
> To enhance the global information in fine-tuning, we introduce perspective-invariant satellite images to establish connections between sparse street-view images. Satellite images are globally consistent and exhibit overlap, allowing for a mapping of the sparse street-view images to a global framework that facilitates inter-image correlations. Previous research in cross-view geo-localization has shown the viability of correlating satellite images with street-view images. In light of this, we propose a method named cross-view alignment tuning, designed to enable LVLMs to align street-view images with street addresses on satellite images annotated with street name labels.
>
> We are sorry that it is difficult to describe the above motivation through theoretical analysis or mathematical derivation. After all, there is no theoretical or mathematical derivation for the research work of cross-view matching. This cross-view approach has been continuously verified in practice and in the work of predecessors.
>
> * Limitation
>
> Our method may face challenges in scenarios involving low-light conditions and street intersections. We have present some failure cases in Fig. 11 of Appendix G. For cases under low-light conditions like tunnels, there is a lack of recognizable features like large areas of sky, walls, and vegetation. For cases near street intersections, the proposed model tends to mistakenly identify them as regular street views or incorrectly recognize them as street names with road signs.
>
>
> **Response to weakness 2**: We have provided a demonstration of the scalability of our method on a dataset with two cities by mixing Pitts-IAL and SF-IAL-Base in Tab. 4. It can be observed that when the dataset size is doubled, our approach is able to maintain performance close to that of training on a single dataset.
>
> **Response to question 1**: Thanks for your suggestions! In the introduction and Fig. 1, we reviewed the related work AddressCLIP and GeoReasoner.
> The former is about the discriminative address localization with CLIP. Although the authors of AddressCLIP presented an experiment using VQA data, they just used a single round of fixed question template (i.e., Where might this photo have been taken? Tell me its street level address.) and only provide qualitative results. In contrast, we focus on the open-set problem and utilize multi-round VQA data with various complex question templates. Besides, we provide detailed quantitative experimental results in terms of various metrics.
> The latter is designed for coarse-grained country and city levels of address localization while ours focuses on fine-grained street level address localization within a city.
>
> **Response to question 2**: In traditional Visual Place Recognition, both the query images and the database images consist of ground-level street views. Consequently, the core objective is to learn a robust feature extraction model that maps images of the same location to the same position in the feature space. However, in Cross-view Geo-localization, the query images are ground-level street views, while the database images are aerial perspectives, such as bird's-eye views. The central goal is to learn a feature extractor that can map these cross-perspective images of the same location to the same position in the feature space. Therefore, a sufficiently powerful feature extraction model may be well-suited for Visual Place Recognition but may not be directly equipped to handle Cross-view Geo-localization.
>
> **Response to question 3**: We clarify that the proposed grafting mechanism is only formally similar to CutMix, but not related to data augmentation or the MixUp series of methods. We mentioned CutMix in the paper only to facilitate readers' understanding of our method, and provided an example in Figure 3. We will remove the confusing expression in our revision.

---

> ### Author Response · Authors · 2024-11-29
> **Looking forward to your reply**
>
> We are very grateful for your feedback and suggestions on our paper and look forward to further discussions with you. Enjoy your holiday!

---

### Official Review · Reviewer_7yPf · 2024-11-02

**Soundness:** 3
**Presentation:** 4
**Contribution:** 3
**Rating:** 8
**Confidence:** 3

**Summary:**

This paper presents AddressVLM, a model for fine-grained geo-localization consisting of two training stages: cross-view alignment tuning and address localization tuning. The authors also introduce two new VQA datasets adapted from existing address localization datasets from Pittsburgh and San Francisco.

**Strengths:**

S1. This paper is very well-written, and the figures are clear. As someone with little background in image address localization, I appreciate the straightforward presentation and review of related work, including Figure 1 which puts the methods of existing work side-by-side with the method in AddressVLM.

S2. The ablation study is thorough, reporting results on different model variants during each training stage.

**Weaknesses:**

W1. I would have liked to see evaluations on how AddressVLM does on other related geolocalization benchmarks adapted for VQA in the same way that Pitts-VQA and SF-Base-VQA were created. For example, comparing performance on OpenStreetView-5M [1] or Geoguessr data like in [2] would better show how this method specifically improves fine-grained address localization and the side effects it has on other related tasks (e.g., does this method detract from more coarse, global understanding?). I think this type of evaluation, even if it shows that AddressVLM decreases performance on other forms of geolocalization, can only strengthen the papers as it gives a more thorough presentation of what the method can and cannot do.

W2. A more thorough study of vision and language backbones would establish a more compelling case for using CLIP and Phi-3.1-mini. SigLIP has been shown to perform better on many VQA tasks, and DINOv2 has been shown to better localize objects in an image (rather than just capture more global, image-level semantics like CLIP); both of these abilities could translate to improvements in the VLM downstream.

[1] Astruc, Guillaume et al. “OpenStreetView-5M: The Many Roads to Global Visual Geolocation.” 2024 IEEE/CVF Conference on Computer Vision and Pattern Recognition (CVPR) (2024): 21967-21977.

[2] Haas, Lukas et al. “PIGEON: Predicting Image Geolocations.” 2024 IEEE/CVF Conference on Computer Vision and Pattern Recognition (CVPR) (2023): 12893-12902.

**Questions:**

Q1. What are some limitations of AddressVLM? Are there particular categories of edge cases (e.g., bias toward one street vs. the other when localizing an intersection, low-light conditions, occlusions) that the method performs particularly poorly at, and are these failure patterns similar to the failure patterns of existing models?

---

> ### Author Response · Authors · 2024-11-20
>
> Thank you very much for the constructive feedback! Below, we address the concerns raised in the review and will adjust the paper accordingly in the revision.
>
> **Response to weakness 1**: Thanks for the valuable suggestion. We appreciate the reviewer for providing the benchmark reference for a comprehensive presentation. However, the two benchmarks, OpenStreetView-5M [1] or Geoguessr [2], cover  coarse-grained address localization of countries and cities around the world, while our work focuses on fine-grained address localization of urban streets and district. These two benchmarks do not match our research problem. Nevertheless, we adopt another address localization benchmark, the Tokyo dataset [Ref 1], for experiments. Since our method includes a series of steps such as address crawling and filtering, satellite image crawling, cross-view alignment label generation, and model fine-tuning, we have not yet obtained experimental results. We believe that AddressVLM will achieve consistent good performances. Once the experiments are completed, we will add the results into the final version.
>
> [Ref 1] 24/7 place recognition by view synthesis. In CVPR 2015.
>
>
> **Response to weakness 2**: Thanks for the valuable suggestion. The vision encoders and LLMs included in current pre-trained LVLMs are typically paired in a rigid manner. Simply replacing CLIP with SigLIP or DINOv2 means we have to retrain the entire LVLM from scratch, which is not the focus of our work and is not realistic for implementations. As an alternative, we conducted a similar experiment using MiniCPM-v2.6, which features a vision encoder of SigLIP and an LLM of Qwen2-7B. The results on Pitts-VQA are shown in the table below.
>
> | Method | $A_d^G$ |  $A_d^J$ |  $A_d^M$ |  $\bar{A}_d$ | $A_s^G$ |  $A_s^J$ |  $A_s^M$ |  $\bar{A}_s$ | $\bar{A}$ | $A_{sd}$ |
> | --- | --- | --- | --- | --- | --- | --- | --- | --- | --- | --- |
> | CLIP + Phi3-mini | 88.73 | 93.54 | 95.16 | 92.70 | 72.51 | 91.70 | 93.98 | 87.46 | 90.02 | 69.60 |
> | SigLIP + Qwen2-7B | 89.82 | 94.21 | 95.85 | 93.57 | 73.69 | 92.73 | 94.52 | 88.58 | 90.97 | 70.49 |
>
> It can be observed that the performance of AddressVLM based on SigLIP and Qwen2-7B is slightly higher than that based on LLaVA-phi3, which demonstrates that adopting a more powerful LVLM can achieve better results. We will add the results in the revised version.
>
>
> **Response to question 1**:  We have present some failure cases in Fig. 11 of Appendix G in the original manuscript. Our model often faces challenges in scenarios involving low-light conditions, such as tunnels, where there is a lack of recognizable features like large areas of sky, walls, and vegetation. Additionally, when examining cases near street intersections, we've observed that the model tends to mistakenly identify them as regular street views or incorrectly recognize them as street names with road signs.
> The limitations in low-light conditions and at street intersections belong to visual and textual challenges, respectively, which are similar to the failure patterns of existing work like GeoReasoner. The conditions where the images were captured are beyond our control, but we are committed to encouraging the model to think more deeply about cases related to street intersections thus enhancing its ability to locate addresses at intersections.

---

> > ### Comment · Reviewer_7yPf · 2024-11-20
> > **thank you**
> >
> > Thank you for the thorough response to my questions and listed weaknesses. I am more confident in the merits of this paper, and have raised my score.

---

> > > ### Author Response · Authors · 2024-11-21
> > >
> > > Thank you very much for your positive feedback on our response. We are truly grateful for your recognition of our paper and for raising your score. Your constructive suggestions have been invaluable in helping us improve the quality of our work. We also appreciate your thorough and thoughtful review process, which has significantly contributed to the refinement of our research.

---

### Official Review · Reviewer_K9em · 2024-11-03

**Soundness:** 3
**Presentation:** 2
**Contribution:** 2
**Rating:** 5
**Confidence:** 5

**Summary:**

This paper introduces AddressVLM, a novel approach to enhance Large Vision Language Models (LVLMs) for street-level address localization. The proposed cross-view alignment tuning strategy leverages both street-view and satellite imagery to improve the model's understanding of urban spatial relationships. The work addresses a gap in current LVLM capabilities - while they perform well at city/country-level geo-localization, they struggle with precise street-level localization within cities. Also, This paper constructs two street-view VQA datasets based on image address localization datasets from Pittsburgh and San Francisco.

**Strengths:**

1. The motivation is clearly stated.
2. The experimental results show the effectiveness of the proposed method.
3. Creating new datasets for the research community.

**Weaknesses:**

1.The paper uses a pre-trained LVLM to generate textual labels that explain why a street-view image matches a satellite image location. However, any errors or biases in these generated labels become training data for the cross-view alignment tuning stage. These errors could be amplified during the subsequent address localization tuning stage.

2.While improvements are shown, absolute performance is still below specialized discriminative models.

3.Limited evaluation on cities outside the US.

4.While the authors present two novel datasets, the paper would benefit from visual or tabular comparisons highlighting the differences of these two datasets relative to existing ones.

**Questions:**

1. In Figure 3 (b, c), the authors do not introduce what the different colors and lengths represent.

2. What is the purpose of comparing AddressCLIP in Table 1? The experimental results show that the proposed method is not superior to AddressCLIP in terms of the A_ds indicator. Additionally, the authors have not introduced the meaning of the A_ds indicator. Do A_ds and A_sd (referenced in line 364) represent the same meaning?

3. The ablation experiment in Table 3 is quite confusing. Please redesign it to better demonstrate the effectiveness of each module.

---

> ### Author Response · Authors · 2024-11-20
>
> Thank you very much for the constructive feedback! Below, we address the concerns raised in the review and will adjust the paper accordingly in the revision.
>
> **Response to weakness 1**: We ensure the accuracy of alignment labels in two ways. 1) As mentioned in lines 261-265 of our paper, the precise location of the street-view image is provided to the pre-trained LVLM in advance (hint: {near Forbes Avenue, Downtown, Pittsburgh}). This ensures that the model is explaining the reasons for cross-view alignment rather than predicting the reasons, thus avoiding potential errors or biases. 2) We manually check the image descriptions and alignment reasons to ensure that they are correct.
> As a result, compared to the baseline of directly applying localization tuning, the proposed alignment tuning stage brings significant and consistent performance gains, e.g., +9.08% and +11.83%, on Pitts-VQA and SF-Base-VQA, respectively.
>
> **Response to weakness 2**: In this paper, our goal is to train a model on question-answer pairs under an open problem setting so that it can answer address-related questions in a more interactive way during the test phase. Although our method is slightly weaker than specific discriminative models with predefined address labels, our method has the following advantages: 1) Open problem setting; 2) Easy to interact with address-related question-answering format; 3) General question-answering capabilities. The key contribution of our method is integrating city-wide address localization capabilities into LVLMs, which provides a promising way of thinking how to learn address-related information from street-view images. We believe this paradigm has the potential to outperform discriminative modeling approaches, for example through larger LVLMs or exploring better tuning approaches.
>
>
> **Response to weakness 3**: Our approach is applicable to modern cities based on administrative address divisions, so we believe that AddressVLM will achieve consistent performances for cities outside the US, regardless of the country in which the city is located. We evaluated AddressVLM on two cities in US for two reasons. 1) Both datsets are widely used in the image geo-localization task, which is convenient for comparison. 2) Our meta address information comes from the AddressCLIP, where only Pittsburgh and San Francisco in the US are covered.
> We agree that evaluating on wider and diverse urban scenarios helps fully demonstrate the effectiveness of our method. We are currently constructing data and validating our method on a completely different localization dataset, the Tokoy dataset [Ref 1], which covers Tokyo, Japan. Since our method includes a series of steps such as address crawling and filtering, satellite image crawling, cross-view alignment label generation, and model fine-tuning, we have not yet obtained experimental results. Once the experiments are completed, we will add the results into the revised version.
>
> [Ref 1] 24/7 place recognition by view synthesis. In CVPR 2015.
>
>
> **Response to weakness 4**: Thanks for the valuable suggestion. We provide the details of the proposed Pitts-VQA and SF-Base-VQA datasets in Table.6 in Appendix. B. A preview of the tabular comparisons with existing related datasets is given below.
>
> | Statistics | Pitts-VQA | SF-Base-VQA | Pitts-IAL | SF-IAL-Base | Pitts-250K | SF-XL |
> | --- | --- | --- | --- | --- | --- | --- |
> | Images | 178K / 19K | 143K / 21K | 234K / 19K | 184K / 21K | 250K / 24K | 41.2M / 1K |
> | GPS | Yes | Yes | Yes | Yes | Yes | Yes|
> | Address | Yes | Yes | Yes | Yes | No | No |
> | QA | 533K / 168K | 430K / 182K | N/A | N/A | N/A | N/A |
>
> We use the training and testing sets in the Pitts-IAL and SF-Base-IAL datasets to construct VQA data, and use them as the training and testing sets of the Pitts-VQA and SF-Base-VQA datasets, respectively, without including the database used for the VPR task in the original datasets. We will add the table of dataset comparisons in the revised version.

---

> > ### Author Response · Authors · 2024-11-20
> >
> > **Response to question 1**: Thanks for the suggestion. In Fig. 3, we use red, orange, and yellow to represent the top three streets predicted by the model in 100 repeated inferences after alignment fine-tuning, where the length of the color bar represents the number of times the model predicted the corresponding street. We will add the description of color and length to the caption of Fig. 3 in the revised version, and the revised content is as follows:
> >
> > "Qualitative comparisons of the street localization probability distribution before and after cross-view alignment tuning. Red, orange, and yellow represent the top three streets predicted by the model after alignment fine-tuning, where the length of the color bar represents the number of times the model predicts the corresponding street. The predicted streets are clustered and distributed close to the true location after cross-view alignment tuning. The source map can be found here."
> >
> >
> > **Response to question 2**: We provide the $A_{sd}$ result for AddressCLIP mainly for the complemetary of the experimental results. $A_{sd}$ and $A_{ds}$ are the same indicators, both used to investigate the model’s capability to concurrently generate both street and district information. We apologize for writing the same indicator in two forms. We will unify it into $A_{sd}$ in the revised version.
> >
> >
> > **Response to question 3**: Thanks for the valuable suggestion. In Tab. 3, we demonstrate the performance impact of freezing different model modules at different tuning stages. The confusion about Tab. 3 may come from the redundant information in the table. We have redesigned the following table to clearly show the ablation settings and results:
> >
> > | Variants | VE-stage1 | LLM-stage1 | VE-stage2 | Pitts-VQA $\bar{A}$ | Pitts-VQA $A_{sd}$ | SF-Base-VQA $\bar{A}$ | SF-Base-VQA $A_{sd}$ |
> > | --- | --- | --- | --- | --- | --- | --- | --- |
> > | A | N | N | N | 86.58 | 63.21 | 86.74 | 62.94 |
> > | B | N | N | Y | 86.42 | 62.95 | 86.31 | 62.78 |
> > | C | N | Y | N | 87.48 | 63.03 | 85.92 | 61.21 |
> > | D | N | Y | Y | 89.53 | 66.37 | 89.63 | 68.95 |
> > | E | Y | Y | N | 87.37 | 63.52 | 87.07 | 64.68 |
> > | AddressVLM | Y | Y | Y | 90.02 | 69.60 | 89.33 | 70.45 |
> >
> > As seen, unfreezing the LLM in stage1 yields the most substantial performance improvement (CD vs AB).  Similarly, unfreezing the VE in stage2 usually achieves better performance than freezing the VE, since the target of the second stage training is street-view images and unfreezing the VE enables the model to better adapt to urban street scenes (A vs B, C vs D). Ultimately, unfreezing all parameters leads to the best performance. We will modify Tab. 3 in the revised version according to the above table.

---

> ### Author Response · Authors · 2024-11-29
> **Looking forward to your reply**
>
> We are very grateful for your feedback and suggestions on our paper and look forward to further discussions with you. Enjoy your holiday!

---

> ### Comment · Reviewer_K9em · 2024-11-30
> **Re-response**
>
> Thanks to the authors for responding to my questions and the listed weakness.
>
> Re-response to W1:
> The author claims providing location hints ensures label accuracy. However, while this guarantees correct location information, it doesn't ensure accurate visual descriptions or alignment reasoning from the LVLM, which may still generate inaccurate or biased feature descriptions.
> Using performance improvements (+9.08% and +11.83%) to validate label quality is logically flawed. The gains could stem from other factors like the cross-view alignment mechanism itself, rather than label quality.
>
> Re-response to W2:
>  "Open problem setting" is incorrectly presented as an advantage. It actually makes the task more challenging and explains the lower performance compared to discriminative models, but shouldn't be listed as a benefit.
> The claim that "this paradigm has the potential to outperform discriminative modeling approaches" lacks substantive evidence. Simply mentioning larger models and better tuning is insufficient - I hope the authors explain why generative approaches could theoretically surpass discriminative methods.
>
> Re-response to W3:
> The response has several questionable claims and statements:
> "Our approach is applicable to modern cities based on administrative address divisions" – There maybe different addressing systems for the world's diverse administrative divisions. This claim lacks evidence since the authors haven't tested on cities with different addressing systems.
> "AddressVLM will achieve consistent performances for cities outside the US" - Making this prediction without enough evidence is not convincing.
> The authors mention ongoing work with Tokyo dataset but provide no concrete timeline or preliminary results, making it unclear if this will actually materialize.

---

> > ### Author Response · Authors · 2024-12-02
> > **Re-response to reviewer  K9em**
> >
> > Thanks for your feedback and further discussion. Below, we address the further concerns and will adjust the paper accordingly in the revision.
> >
> > **Re-response to W1**: While we have manually reviewed the description labels from the stage 1 of alignment tuning, we acknowledge that this alone cannot completely eliminate the risk of errors and biases introduced by hallucinations of LVLMs. Hallucinations remain a common challenge with current LLMs, and this issue is frequently encountered in methodologies that leverage LLM or LVLM for automatic label generation. Thus, we respectfully disagree that this is a shortcoming of our approach. Conversely, we have demonstrated the feasibility and advantages of our method using the open-source LLaVA-v1.6-Mistral-7B. If sufficient funding and resources become available, utilizing GPT-4o for the stage 1 of label generation would be ideal.
> >
> > **Re-response to W2**: Thank you for acknowledging our focus on a more challenging problem. We emphasize that, compared to the discriminative model, our approach allows for more flexible question answering at test time, offers enhanced interactivity, and does not require pre-defined addresses. As you noted, under such an open problem setting, the challenges and complexities we need to address in this study are elevated to a higher level. Consequently, the performance of our method does not currently match that of the discriminative model.
> >
> > We apologize for our previous assumption that the generative model could potentially surpass the discriminative model in performance. In reality, for many tasks (such as visual grounding, depth estimation, etc.), the generative paradigm-based LVLM does not yet achieve the same results as proprietary models based on the discriminative paradigm. However, these discrepancies serve as the driving force for us to explore and develop more flexible and interactive methods. This pursuit defines the value of our research and marks a progression beyond previous discriminative models. We remain committed to narrowing the performance gap between generative and discriminative models in our future work.
> >
> > **Re-response to W3**: There are indeed different administrative address division methods and address systems in the world, but their commonality is that addresses are hierarchical. For instance, addresses in the UK are structured into Town/Village and Streets, those in Japan into Machi, Chome, and Streets, while in France and India, they primarily consist of Streets. Despite varying in descriptions and levels of granularity, all address systems in cities can be generally broken down into 1-3 levels. By gathering and constructing address-related VQA data tailored to the hierarchical structures of different cities, our method can be effectively applied to modern urban areas.
> >
> > Regarding our experiments on the Tokyo dataset, we have been diligently engaged in data collection, annotation, training, and evaluation since the rebuttal process began. The numerous steps involved made it challenging to predict the timeline for completing the experiments during the early stages of the rebuttal. Fortunately, with the extension of the rebuttal deadline, we were able to finalize the experiment. The detailed process of our experiment is as follows.
> >
> > In the Tokyo dataset related to image geolocation tasks, we collected a total of  52080 street view images (from 4340 locations with 12 images for each location), along with 52764 sets of VQA dialogue data and corresponding satellite images for training. The addresses were categorized into two levels: Chome and Street. We completed two stages of fine-tuning, and the results on the test set (7440 images from 1240 locations) are as follows:
> >
> > As can be seen, our method demonstrated good performance on the Tokyo dataset even if the addresses are highly biased, effectively utilizing Tokyo's address division system. This underscores the method's scalability and adaptability in modern cities with the well-defined address-related VQA data.
> > | Method | $A_d^G$ |  $A_d^J$ |  $A_d^M$ |  $\bar{A}_d$ | $A_s^G$ |  $A_s^J$ |  $A_s^M$ |  $\bar{A}_s$ | $\bar{A}$ | $A_{sd}$ |
> > | --- | --- | --- | --- | --- | --- | --- | --- | --- | --- | --- |
> > | AddressVLM | 73.85 | 89.49 | 88.62 | 84.99 | 63.52 | 88.28 | 86.03 | 81.57 | 82.28 | 65.81 |
> > | AddressVLM w/o stage1 | 70.63 | 86.22 | 85.93 | 83.14 | 58.06 | 85.39 | 84.72 | 78.37 | 80.76 | 56.37 |

---

### Meta-Review · Area_Chair_yL7c · 2024-12-18

**Metareview:**

To enhance LVLMs for fine-grained street-level localization in urban areas, this paper proposed AddressVLM which consists of two training stages: cross-view alignment tuning and address localization. The main innovations of AddressVLM include exploring the integration of address localization capability within a city into LVLM, proposing a cross-view alignment fine-tuning method, and introducing the AddressVLM model. The advantage of the paper lies in its clear motivation, creating new datasets and the ability to achieve achieve . The weaknesses mainly include lack of theoretical analysis and mathematical derivation , limited evaluation on cities outside the US, lack of discussions on impacts of errors or biases in the generated labels. I think the shortcomings of the paper outweigh its advantages and recommend reject to this paper.

**Additional Comments On Reviewer Discussion:**

The concerns raised by reviewers focus on quality of generated labels, limited evaluation, unclear descriptions, insufficient comparison with related works, lack of theoretical analysis, analysis of impact of language backbones and hyper-parameters and quantitative experiments, etc. The authors provided lots of explanations and some experimental results. However, concern about “limited evaluation on cities outside the US”,  “lack of discussions on impacts of errors or biases in the generated labels” and “lack of comparison with  traditional geolocation and geolocation methods” remains unresolved. Three reviewers maintained their ratings as “marginally below the acceptance threshold”after rebuttal. Therefore, I recommend another round of revision to incorporate the excellent suggestions provided by the insightful reviewers.

---

### Decision · Program_Chairs · 2025-01-22

Reject